# Accurate and versatile 3D segmentation of plant tissues at cellular resolution

**Adrian Wolny[1,2†], Lorenzo Cerrone[1†], Athul Vijayan[3], Rachele Tofanelli[3], Amaya Vilches Barro[4], Marion Louveaux[4‡], Christian Wenzl[4], Sören Strauss[5], David Wilson-Sánchez[5], Rena Lymbouridou[5], Susanne S Steigleder[4], Constantin Pape[1,2], Alberto Bailoni[1], Salva Duran-Nebreda[6], George W Bassel[6], Jan U Lohmann[4], Miltos Tsiantis[5], Fred A Hamprecht[1], Kay Schneitz[3], Alexis Maizel[4], Anna Kreshuk[2]***

[1]Heidelberg Collaboratory for Image Processing, Heidelberg University, Heidelberg, Germany; [2]EMBL, Heidelberg, Germany; [3]School of Life Sciences Weihenstephan, Technical University of Munich, Freising, Germany; [4]Centre for Organismal Studies, Heidelberg University, Heidelberg, Germany; [5]Department of Comparative Development and Genetics, Max Planck Institute for Plant Breeding Research, Cologne, Germany; [6]School of Life Sciences, University of Warwick, Coventry, United Kingdom

**Abstract** Quantitative analysis of plant and animal morphogenesis requires accurate segmentation of individual cells in volumetric images of growing organs. In the last years, deep learning has provided robust automated algorithms that approach human performance, with applications to bio-image analysis now starting to emerge. Here, we present PlantSeg, a pipeline for volumetric segmentation of plant tissues into cells. PlantSeg employs a convolutional neural network to predict cell boundaries and graph partitioning to segment cells based on the neural network predictions. PlantSeg was trained on fixed and live plant organs imaged with confocal and light sheet microscopes. PlantSeg delivers accurate results and generalizes well across different tissues, scales, acquisition settings even on non plant samples. We present results of PlantSeg applications in diverse developmental contexts. PlantSeg is free and open-source, with both a command line and a user-friendly graphical interface.

*For correspondence:
anna.kreshuk@embl.de

†These authors contributed equally to this work

Present address: ‡Institute Pasteur, Paris, France

Competing interests: The authors declare that no competing interests exist.

## Introduction

Large-scale quantitative study of morphogenesis in a multicellular organism entails an accurate estimation of the shape of all cells across multiple specimen. State-of-the-art light microscopes allow for such analysis by capturing the anatomy and development of plants and animals in terabytes of high-resolution volumetric images. With such microscopes now in routine use, segmentation of the resulting images has become a major bottleneck in the downstream analysis of large-scale imaging experiments. A few segmentation pipelines have been proposed (*Fernandez et al., 2010*; *Stegmaier et al., 2016*), but these either do not leverage recent developments in the field of computer vision or are difficult to use for non-experts.

With a few notable exceptions, such as the Brainbow experiments (*Weissman and Pan, 2015*), imaging cell shape during morphogenesis relies on staining of the plasma membrane with a fluorescent marker. Segmentation of cells is then performed based on their boundary prediction. In the early days of computer vision, boundaries were usually found by edge detection algorithms (*Canny, 1986*). More recently, a combination of edge detectors and other image filters was commonly used as input for a machine learning algorithm, trained to detect boundaries (*Lucchi et al., 2012*). Currently, the most powerful boundary detectors are based on Convolutional Neural

Networks (CNNs) (*Long et al., 2015*; *Kokkinos, 2015*; *Xie and Tu, 2015*). In particular, the U-Net architecture (*Ronneberger et al., 2015*) has demonstrated excellent performance on 2D biomedical images and has later been further extended to process volumetric data (*Çiçek et al., 2016*).

Once the boundaries are found, other pixels need to be grouped into objects delineated by the detected boundaries. For noisy, real-world microscopy data, this post-processing step still represents a challenge and has attracted a fair amount of attention from the computer vision community (*Turaga et al., 2010*; *Nunez-Iglesias et al., 2014*; *Beier et al., 2017*; *Wolf et al., 2018*; *Funke et al., 2019a*). If centroids ('seeds') of the objects are known or can be learned, the problem can be solved by the watershed algorithm (*Couprie et al., 2011*; *Cerrone et al., 2019*). For example, in *Eschweiler et al., 2018* a 3D U-Net was trained to predict cell contours together with cell centroids as seeds for watershed in 3D confocal microscopy images. This method, however, suffers from the usual drawback of the watershed algorithm: misclassification of a single cell centroid results in sub-optimal seeding and leads to segmentation errors.

Recently, an approach combining the output of two neural networks and watershed to detect individual cells showed promising results on segmentation of cells in 2D (*Wang et al., 2019*). Although this method can in principle be generalized to 3D images, the necessity to train two separate networks poses additional difficulty for non-experts.

While deep learning-based methods define the state-of-the-art for all image segmentation problems, only a handful of software packages strives to make them accessible to non-expert users in biology (reviewed in [*Moen et al., 2019*]). Notably, the U-Net segmentation plugin for ImageJ (*Falk et al., 2019*) conveniently exposes U-Net predictions and computes the final segmentation from simple thresholding of the probability maps. CDeep3M (*Haberl et al., 2018*) and DeepCell (*Van Valen et al., 2016*) enable, via the command-line, the thresholding of the probability maps given by the network, and DeepCell allows instance segmentation as described in *Wang et al., 2019*. More advanced post-processing methods are provided by the ilastik Multicut workflow (*Berg et al., 2019*), however, these are not integrated with CNN-based prediction.

Here, we present PlantSeg, a deep learning-based pipeline for volumetric instance segmentation of dense plant tissues at single-cell resolution. PlantSeg processes the output from the microscope with a CNN to produce an accurate prediction of cell boundaries. Building on the insights from previous work on cell segmentation in electron microscopy volumes of neural tissue (*Beier et al., 2017*; *Funke et al., 2019a*), the second step of the pipeline delivers an accurate segmentation by solving a graph partitioning problem. We trained PlantSeg on 3D confocal images of fixed *Arabidopsis thaliana* ovules and 3D+t light sheet microscope images of developing lateral roots, two standard imaging modalities in the studies of plant morphogenesis. We investigated a range of network architectures and graph partitioning algorithms and selected the ones which performed best with regard to extensive manually annotated ground truth. We benchmarked PlantSeg on a variety of datasets covering a range of samples and image resolutions. Overall, PlantSeg delivers excellent results on unseen data and, as we show through quantitative and qualitative evaluation, even non-plant datasets do not necessarily require network retraining. Combining the repository of accurate neural networks trained on the two common microscope modalities and going beyond just thresholding or watershed with robust graph partitioning strategies is the main strength of our package. PlantSeg is an open-source tool which contains the complete pipeline for segmenting large volumes. Each step of the pipeline can be adjusted via a convenient graphical user interface while expert users can modify configuration files and run PlantSeg from the command line. Users can also provide their own pre-trained networks for the first step of the pipeline using a popular 3D U-Net implementation (https://github.com/wolny/pytorch-3dunet), which was developed as a part of this project. Although PlantSeg was designed to segment 3D images, one can directly use it to segment 2D stacks. Besides the tool itself, we provide all the networks we trained for the confocal and light sheet modalities at different resolution levels and make all our training and validation data publicly available All datasets used to support the findings of this study have been deposited in https://osf.io/uzq3w. All the source code can be found at (*Wolny, 2020a*; https://github.com/hci-unihd/plant-seg; copy archived at https://github.com/elifesciences-publications/plant-seg).

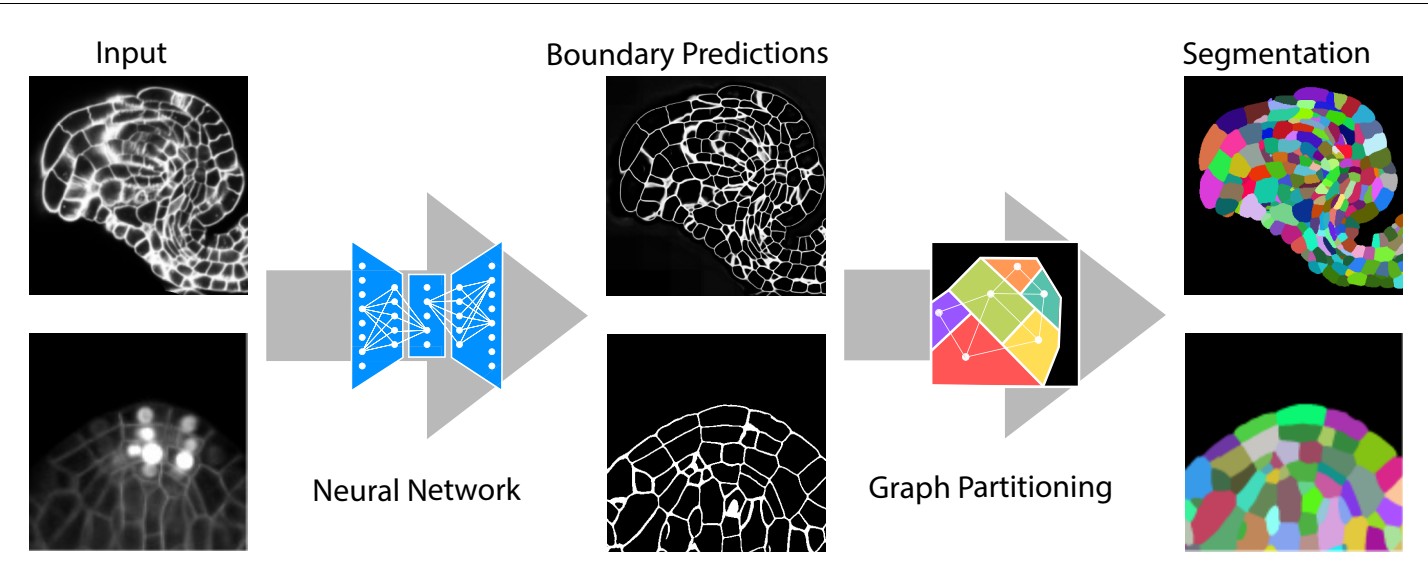

**Figure 1.** Segmentation of plant tissues into cells using PlantSeg. First, PlantSeg uses a 3D UNet neural network to predict the boundaries between cells. Second, a volume partitioning algorithm is applied to segment each cell based on the predicted boundaries. The neural networks were trained on ovules (top, confocal laser scanning microscopy) and lateral root primordia (bottom, light sheet microscopy) of *Arabidopsis thaliana*.

## Results

### A pipeline for segmentation of plant tissues into cells

The segmentation algorithm we propose contains two major steps. In the first step, a fully convolutional neural network (a variant of U-Net) is trained to predict cell boundaries. Afterwards, a region adjacency graph is constructed from the pixels with edge weights computed from the boundary predictions. In the second step, the final segmentation is computed as a partitioning of this graph into an unknown number of objects (see *Figure 1*). Our choice of graph partitioning as the second step is inspired by a body of work on segmentation for nanoscale connectomics (segmentation of cells in electron microscopy images of neural tissue), where such methods have been shown to outperform more simple post-processing of the boundary maps (*Beier et al., 2017*; *Funke et al., 2019a*; *Briggman et al., 2009*).

### Datasets

To make our tool as generic as possible, we used both fixed and live samples as core datasets for design, validation and testing. Two microscope modalities common in studies of morphogenesis were employed, followed by manual and semi-automated annotation of ground truth segmentation.

The first dataset consists of fixed *Arabidopsis thaliana* ovules at all developmental stages acquired by confocal laser scanning microscopy with a voxel size of 0.075 × 0.075 × 0.235 µm. 48 volumetric stacks with hand-curated ground truth segmentation were used. A complete description of the image acquisition settings and the ground truth creation protocol is reported in *Tofanelli et al., 2019*.

The second dataset consists of three time-lapse videos showing the development of *Arabidopsis thaliana* lateral root primordia (LRP). Each recording was obtained by imaging wild-type Arabidopsis plants expressing markers for the plasma membrane and the nuclei (*Vilches Barro et al., 2019*) using a light sheet fluorescence microscope (LSFM). Stacks of images were acquired every 30 min with constant settings across movies and time points, with a voxel size of 0.1625 × 0.1625 × 0.250 µm. The first movie consists of 52 time points of size 2048 × 1050 × 486 voxels. The second movie consists of 90 time points of size 1940 × 1396 × 403 voxels and the third one of 92 time points of size 2048 × 1195 × 566 voxels. The ground truth was generated for 27 images depicting different developmental stages of LRP coming from the three movies (see Appendix 1 Groundtruth Creation).

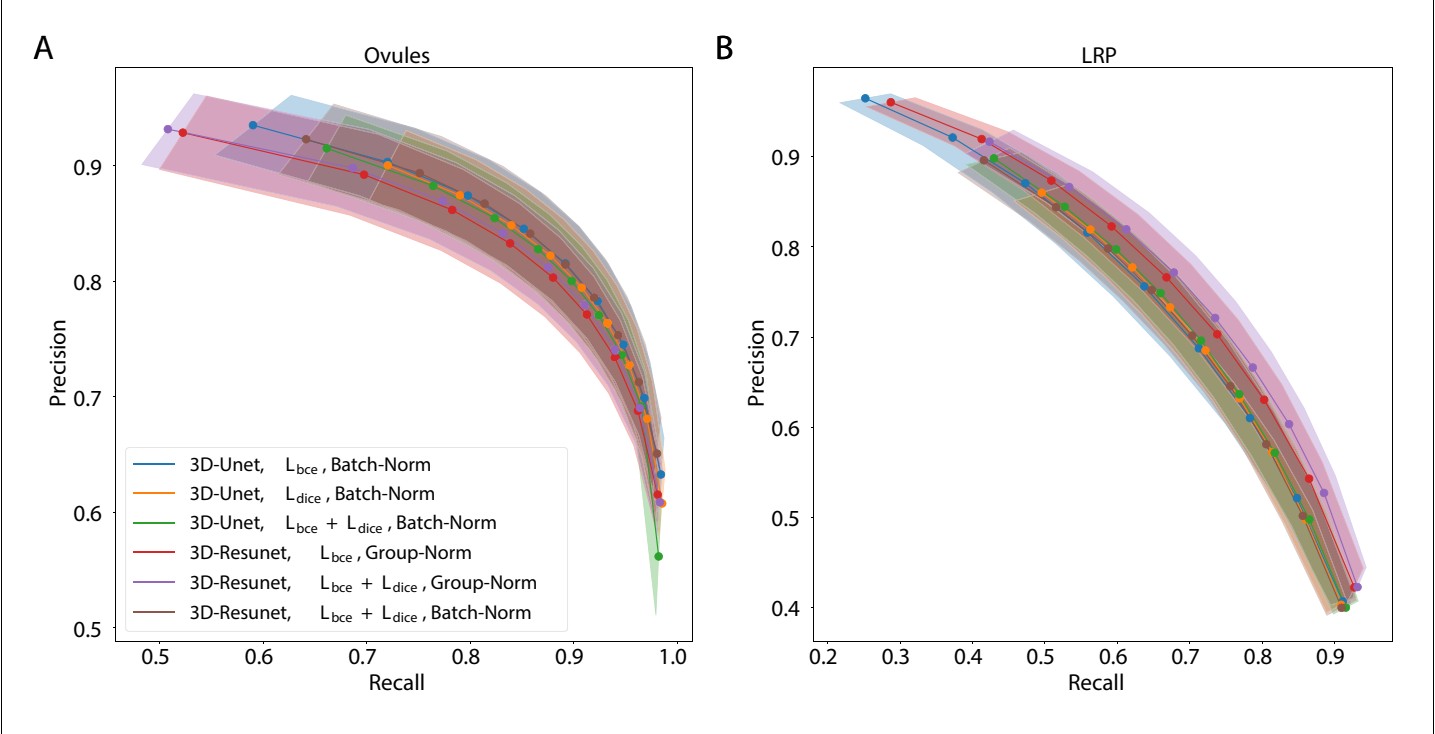

**Figure 2.** Precision-recall curves for different CNN variants on the ovule (**A**) and lateral root primordia (LRP) (**B**) datasets. Six training procedures that sample different type of architecture (3D U-Net *vs.* 3D Residual U-Net), loss function (BCE *vs.* Dice *vs.* BCE-Dice) and normalization (Group-Norm *vs.* Batch-Norm) are shown. Those variants were chosen based on the accuracy of boundary prediction task: three best performing models on the ovule and three best performing models on the lateral root datasets (see *Appendix 5—table 1* for a detailed summary). Points correspond to averages of seven (ovules) and four (LRP) values and the shaded area represent the standard error. For a detailed overview of precision-recall curves on individual stacks we refer to *Appendix 4—figure 1*. Source files used to generate the plot are available in the *Figure 2—source data 1*.

The online version of this article includes the following source data for figure 2:

**Source data 1.** Source data for precision/recall curves of different CNN variants in *Figure 2*.

The two datasets were acquired on different types of microscopes and differ in image quality. To quantify the differences we used the peak signal-to-noise (PSNR) and the structural similarity index measure (SSIM) (*Horé and Ziou, 2010*). We computed both metrics using the input images and their ground truth boundary masks; higher values show better quality. The average PSNR measured on the light sheet dataset was $22.5 \pm 6.5$ dB (average $\pm$ SD), 3.4 dB lower than the average PSNR computed on the confocal dataset ($25.9 \pm 5.7$). Similarly, the average SSIM is $0.53 \pm 0.12$ for the light sheet, 0.1 lower than $0.63 \pm 0.13$ value measured on the confocal images. Both datasets thus contain a significant amount of noise. LSFM images are noisier and more difficult to segment, not only because of the noise, but also due to part of nuclear labels being in the same channel as membrane staining.

In the following we describe in detail the design and performance of each of the two steps of the pipeline.

## Step 1: cell boundary detection

Being the current state of the art in bioimage segmentation, U-Net (*Ronneberger et al., 2015*) was chosen as the base model architecture for predicting the boundaries between cells. Aiming for the best performance across different microscope modalities, we explored various components of neural network design critical for improved generalization and robustness to noise, namely: the network architecture, loss function, normalization layers and size of patches used for training. For the final PlantSeg package we trained one set of CNNs for each dataset as the ovule and lateral root datasets are substantially different.

**Table 1.** Quantification of PlantSeg performance on the 3D Digital Tissue Atlas, using PlantSeg . The Adapted Rand error (ARand) assesses the overall segmentation quality whereas $VOI_{merge}$ and $VOI_{split}$ assess erroneous merge and splitting events. The petal images were not included in our analysis as they are very similar to the leaf and the ground truth is fragmented, making it difficult to evaluate the results from the pipeline in a reproducible way. Segmented images are computed using GASP partitioning with default parameters (left table) and fine-tuned parameters described in Appendix 7: Empirical Example of parameter tuning (right table).

| Dataset | PlantSeg (default parameters) | | | PlantSeg (tuned parameters) | | |
| --- | --- | --- | --- | --- | --- | --- |
| | ARand | $VOI_{split}$ | $VOI_{merge}$ | ARand | $VOI_{split}$ | $VOI_{merge}$ |
| Anther | 0.328 | 0.778 | 0.688 | 0.167 | 0.787 | 0.399 |
| Filament | 0.576 | 1.001 | 1.378 | 0.171 | 0.687 | 0.487 |
| Leaf | 0.075 | 0.353 | 0.322 | 0.080 | 0.308 | 0.220 |
| Pedicel | 0.400 | 0.787 | 0.869 | 0.314 | 0.845 | 0.604 |
| Root | 0.248 | 0.634 | 0.882 | 0.101 | 0.356 | 0.412 |
| Sepal | 0.527 | 0.746 | 1.032 | 0.257 | 0.690 | 0.966 |
| Valve | 0.572 | 0.821 | 1.315 | 0.300 | 0.494 | 0.875 |

In more detail, with regard to the network architecture we compared the regular 3D U-Net as described in *Çiçek et al., 2016* with a Residual U-Net from *Lee et al., 2017*. We tested two loss functions commonly used for the semantic segmentation task: binary cross-entropy (BCE) ($\mathcal{L}_{BCE}$) (*Ronneberger et al., 2015*) and Dice loss ($\mathcal{L}_{Dice}$) (*Sudre et al., 2017*), as well as their linear combination ($\alpha\mathcal{L}_{BCE} + \beta\mathcal{L}_{Dice}$) termed BCE-Dice. The patch size and normalization layers were investigated jointly by comparing training on a single large patch, versus training on multiple smaller patches per network iteration. For single patch we used group normalization (*Wu and He, 2018*) whereas standard batch normalization (*Ioffe and Szegedy, 2015*) was used for the multiple patches.

In the ovule dataset, 39 stacks were randomly selected for training, two for validation and seven for testing. In the LRP dataset, 27 time points were randomly selected from the three videos for training, two time points were used for validation and four for testing.

The best performing CNN architectures and training procedures is illustrated by the precision/recall curves evaluated at different threshold levels of the predicted boundary probability maps (see *Figure 2*). Training with a combination of binary cross-entropy and Dice loss ($L_{bce} + L_{dice}$) performed best on average across the two datasets in question contributing to 3 out of 6 best performing network variants. BCE-Dice loss also generalized well on the out of sample data described in 2.1.4 Performance on external plant datasets. Due to the regularity of cell shapes, the networks do not benefit from broader spatial context when only cell membrane signal is present in input images. Indeed, training the networks with bigger patch sizes does not noticeably increase the performance as compared to training with smaller patches. 4 out of 6 best performing networks use smaller patches and batch normalization (*Batch-Norm*) whereas only 2 out of 6 use bigger patches and group normalization (*Group-Norm*). Residual U-Net architecture (*3D-ResUnet*) performed best on the LRP dataset (*Figure 2B*), whereas standard U-Net architecture (*3D-Unet*) was better on the ovule datasets (*Figure 2A*). For a complete overview of the performance of investigated models see also *Figure 1* and *Appendix 5—table 1*. In conclusion, choosing the right loss function and normalization layers increased the final performance on the task of boundary prediction on both microscope modalities.

## Step 2: segmentation of tissues into cells using graph partitioning

After the cell boundaries are predicted, segmentation of the cells can be formulated as a generic graph partitioning problem, where the graph is built as the region adjacency graph of the image voxels. However, solving the partitioning problem directly at voxel-level is computationally expensive for volumes of biologically relevant size. To make the computation tractable, we first cluster the voxels into so-called supervoxels by running a watershed algorithm on the distance transform of the boundary map, seeded by all local maxima of the distance transform smoothed by a Gaussian blur.

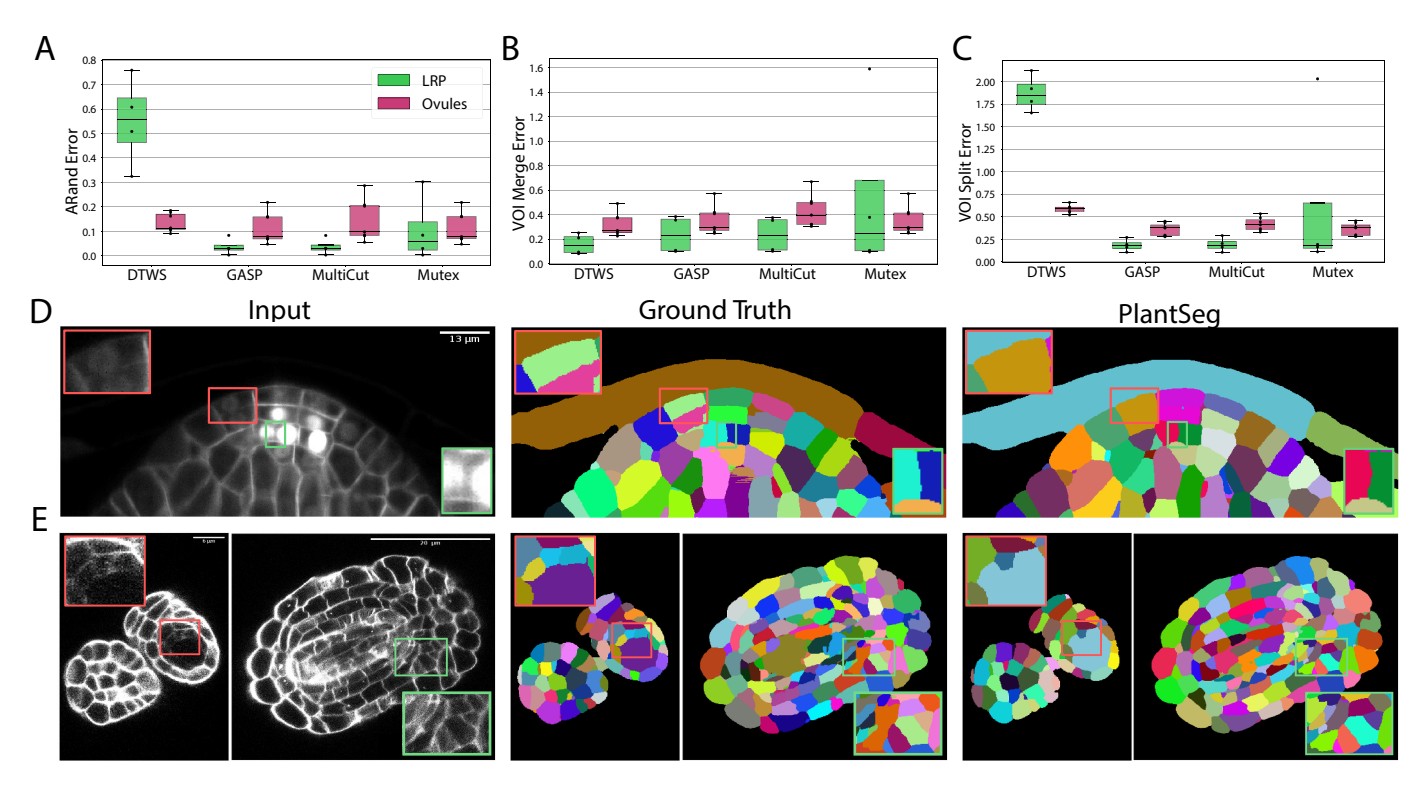

**Figure 3.** Segmentation using graph partitioning. (**A–C**) Quantification of segmentation produced by Multicut, GASP, Mutex watershed (Mutex) and DT watershed (DT WS) partitioning strategies. The Adapted Rand error (**A**) assesses the overall segmentation quality whereas $VOI_{merge}$ (**B**) and $VOI_{split}$ (**C**) assess erroneous merge and splitting events (lower is better). Box plots represent the distribution of values for seven (ovule, magenta) and four (LRP, green) samples. (**D, E**) Examples of segmentation obtained with PlantSeg on the lateral root (**D**) and ovule (**E**) datasets. Green boxes highlight cases where PlantSeg resolves difficult cases whereas red ones highlight errors. We obtained the boundary predictions using the *generic-confocal-3d-unet* for the ovules dataset and the *generic-lightsheet-3d-unet* for the root. All agglomerations have been performed with default parameters. 3D superpixels instead of 2D superpixels were used. Source files used to create quantitative results shown in (**A–C**) are available in the ***Figure 3—source data 1***.
The online version of this article includes the following source data for figure 3:

**Source data 1.** Source data for panes A, B and C in ***Figure 3***.

The region adjacency graph is then built directly on supervoxels and partitioned into an unknown number of segments to deliver a segmentation. We tested four different partitioning strategies: Multicut (***Kappes et al., 2011***), hierarchical agglomeration as implemented in GASP average (GASP) (***Bailoni et al., 2019***), Mutex watershed (Mutex) (***Wolf et al., 2018***) as well as the distance transform (DT) watershed (***Roerdink and Meijster, 2000***) as a baseline since similar methods have been proposed previously (***Eschweiler et al., 2018***; ***Wang et al., 2019***).

To quantify the accuracy of the four segmentation strategies, we use Adapted Rand error (ARand) for the overall segmentation quality and two other metrics based on the variation of information (***Meilă, 2007***) (see Metrics used for evaluation), measuring the tendency to over-split ($VOI_{split}$) or over-merge ($VOI_{merge}$). GASP, Multicut and Mutex watershed consistently produced accurate segmentation on both datasets with low ARand errors and low rates of merge and split errors (***Figure 3A–C*** and ***Appendix 5—table 2***). As expected DT watershed tends to over-segment with higher split error and resulting higher ARand error. Multicut solves the graph partitioning problem in a globally optimal way and is therefore expected to perform better compared to greedy algorithms such as GASP and Mutex watershed. However, in our case the gain was marginal, probably due to the high quality of the boundary predictions.

The performance of PlantSeg was also assessed qualitatively by expert biologists. The segmentation quality for both datasets is very satisfactory. For example in the lateral root dataset, even in cases where the boundary appeared masked by the high brightness of the nuclear signal, the

network correctly detected it and separated the two cells (*Figure 3D*, green box). On the ovule dataset, the network is able to detect weak boundaries and correctly separate cells in regions where the human expert fails (*Figure 3E* , green box). The main mode of error identified in the lateral root dataset is due to the ability of the network to remove the nuclear signal which can weaken or remove part of the adjacent boundary signal leading to missing or blurry cell contour. For example, the weak signal of a newly formed cell wall close to two nuclei was not detected by the network and the cells were merged (*Figure 3D*, red box). For the ovule dataset, in rare cases of very weak boundary signal, failure to correctly separate cells could also be observed (*Figure 3E*, red box).

Taken together, our analysis shows that segmentation of plant tissue using graph partitioning handles robustly boundary discontinuities present in plant tissue segmentation problems.

## Performance on external plant datasets

To test the generalization capacity of PlantSeg, we assessed its performance on data for which no network training was performed. To this end, we took advantage of the two publicly available datasets: Arabidopsis 3D Digital Tissue Atlas (https://osf.io/fzr56) composed of eight stacks of eight different *Arabidopsis thaliana* organs with hand-curated ground truth (*Bassel, 2019*), as well as the developing leaf of the Arabidopsis (*Fox et al., 2018*) with 3D segmentation given by the SimpleITK package (*Lowekamp et al., 2013*). The input images from the digital tissue atlas are confocal stacks of fixed tissue with stained cell contours and thus similar to the images of the Arabidopsis ovules, whereas the images of the leaf were acquired through the use of live confocal imaging. It's important to note that the latter image stacks contain highly lobed epidermal cells, which are difficult to segment with classical watershed-based algorithms. We fed the confocal stacks to PlantSeg and qualitatively assessed the resulting segmentation. Quantitative assessment was performed only for the digital tissue atlas, where the ground truth labels are available.

Qualitatively, PlantSeg performed well on both datasets, giving satisfactory results on all organs from the 3D Digital Tissue Atlas, correctly segmenting even the oval non-touching cells of the anther and leaf: a cell shape not present in the training data (*Figure 4*). Our pipeline yielded especially good segmentation results when applied to the complex epidermal cells, visibly outperforming the results obtained using the SimpleITK framework (*Figure 5*).

Quantitatively, the performance of PlantSeg out of the box (default parameters) on the 3D Digital Tissue Atlas is on par with the scores reported on the LRP and ovules datasets on the anther, leaf, and the root, but lower for the other tissues (*Table 1*, left). Default parameters have been chosen to deliver good results on most type of data, however we show that a substantial improvement can be obtained by parameter tuning (see Appendix 6: PlantSeg - Parameters Guide for an overview of the pipeline's hyperparameters and Appendix 7: Empirical Example of parameter tuning for a detailed guide on empirical parameter tuning), in case of the tissue 3D Digital Tissue Atlas tuning improved segmentation by a factor of two as measured with the ARand error (*Table 1*, right). It should be noted that the ground truth included in the dataset was created for analysis of the cellular connectivity network, with portions of the volumes missing or having low quality ground truth (see e.g filament and sepal in *Figure 4*). For this reason, the performance of PlantSeg on these datasets may be underestimated.

Altogether, PlantSeg performed well qualitatively and quantitatively on datasets acquired by different groups, on different microscopes, and at different resolutions than the training data. This demonstrates the generalization capacity of the pre-trained models from the PlantSeg package.

## Performance on a non-plant benchmark

For completeness, we compared PlantSeg performance with state-of-the-art methods on a non-plant open benchmark consisting of epithelial cells of the *Drosophila* wing disc (*Funke et al., 2019b*). Visually, the benchmark images are quite similar to the ovules dataset: membrane staining is used along with a confocal microscope, and the cell shapes are compact and relatively regular. Although we did not train the neural networks on the benchmark datasets and used only the ovule pre-trained models provided with the PlantSeg package, we found out that PlantSeg is very competitive qualitatively and quantitatively and faster than the state-of-the-art methods, all of which rely on networks trained directly on the benchmark dataset (see Appendix 3: Performance of PlantSeg on an independent reference benchmark for detailed overview of the benchmark results). We argue that the large selection

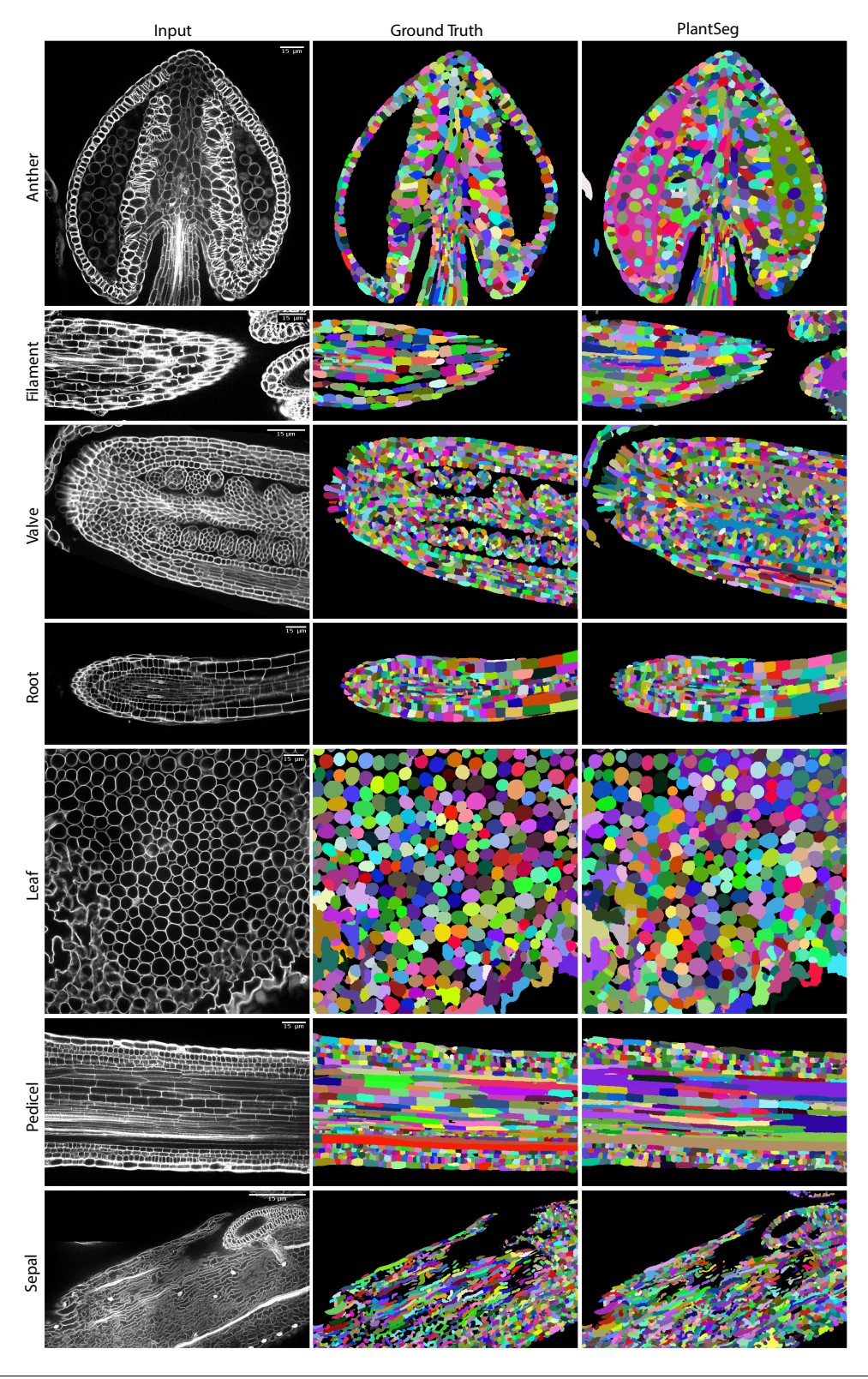

**Figure 4.** PlantSeg segmentation of different plant organs of the 3D Digital Tissue Atlas dataset, not seen in training. The input image, ground truth and segmentation results using PlantSeg are presented for each indicated organ.

of pre-trained networks and graph partitioning algorithms make PlantSeg versatile enough to work well on wide variety of membrane stained tissues, beyond plant samples.

## A package for plant tissue segmentation and benchmarking

We release PlantSeg as an open-source software for the 2D and 3D segmentation of cells with cell contour staining. PlantSeg allows for data pre-processing, boundary prediction with neural networks and segmentation of the network output with a choice of four partitioning methods: Multicut, GASP, Mutex watershed and DT watershed.

PlantSeg can be executed via a simple graphical user interface or via the command line. The critical parameters of the pipeline are specified in a configuration file. Running PlantSeg from the graphical user interface is well suited for processing of single experiments, while the use of the command line utility enables large scale batch processing and remote execution. Our software can export both the segmentation results and the boundary probability maps as Hierarchical Data Format (HDF5) or Tagged Image File Format (TIFF). Both file formats are widely supported and the results can be further processed by other bioimage analysis tools, such as ilastik, MorphographX or Fiji. In particular: the final segmentation is exported as a labeled volume where all pixels of each segmented cell are assigned the same integer value. It is best viewed with a random lookup table, such as 'glasbey' in Fiji. In exported boundary probability maps each pixel has a floating point number between 0 and 1 reflecting a probability of that pixel belonging to a cell boundary. PlantSeg comes with several 2D and 3D networks pre-trained on different voxel size of the Arabidopsis ovule and LRP datasets. Users can select from the available set of pre-trained networks the one with features most similar to their datasets. Alternatively, users can let PlantSeg select the pre-trained network based on the microscope modality (light sheet or confocal) and voxel size. PlantSeg also provides a command-line tool for training the network on new data when none of the pre-trained network is suitable to the user's needs.

PlantSeg is publicly available https://github.com/hci-unihd/plant-seg. The repository includes a complete user guide and the evaluation scripts used for quantitative analysis. Besides the source code, we provide a Linux conda package and a docker image which allows to run PlantSeg on non-Linux operating systems. The software is written in Python, the neural networks use the PyTorch framework *Paszke et al., 2019*. We also make available the raw microscopic images as well as the ground truth used for training, validation and testing.

## Applications of PlantSeg

Here, we show case applications of PlantSeg to the analysis of the growth and differentiation of plant organs at cellular resolution.

## Variability in cell number of ovule primordia

Ovule development in *Arabidopsis thaliana* has been described to follow a stereotypical pattern (*Robinson-Beers et al., 1992*; *Schneitz et al., 1995*). However, it is unclear if ovules within a pistil develop in a synchronous fashion.

Taking advantage of PlantSeg we undertook an initial assessment of the regularity of primordia formation between ovules developing within a given pistil (*Figure 6*). We noticed that spacing between primordia is not uniform (*Figure 6A*). Our results further indicated that six out of the eight analyzed stage 1 primordia (staging according to *Schneitz et al., 1995*) showed a comparable number of cells (140.5 ± 10.84, mean ± SD, ovules 1–5, 7) (*Figure 6B*). However, two primordia exhibited a smaller number of cells with ovule #6 being composed of 91 and ovule #8 of 57 cells, respectively. Interestingly, we observed that the cell number of a primordium does not necessarily translate into its respective height or proximal-distal (PD) extension. For example, ovule #2, which is composed of 150 cells and thus of the second largest number of cells of the analyzed specimen, actually represents the second shortest of the eight primordia with a height of $26.5 \mu m$ (*Figure 6C*). Its comparably large number of cells relates to a wide base of the primordium. Taken together, this analysis indicates that ovule primordium formation within a pistil is relatively uniform, however, some variability can be observed.

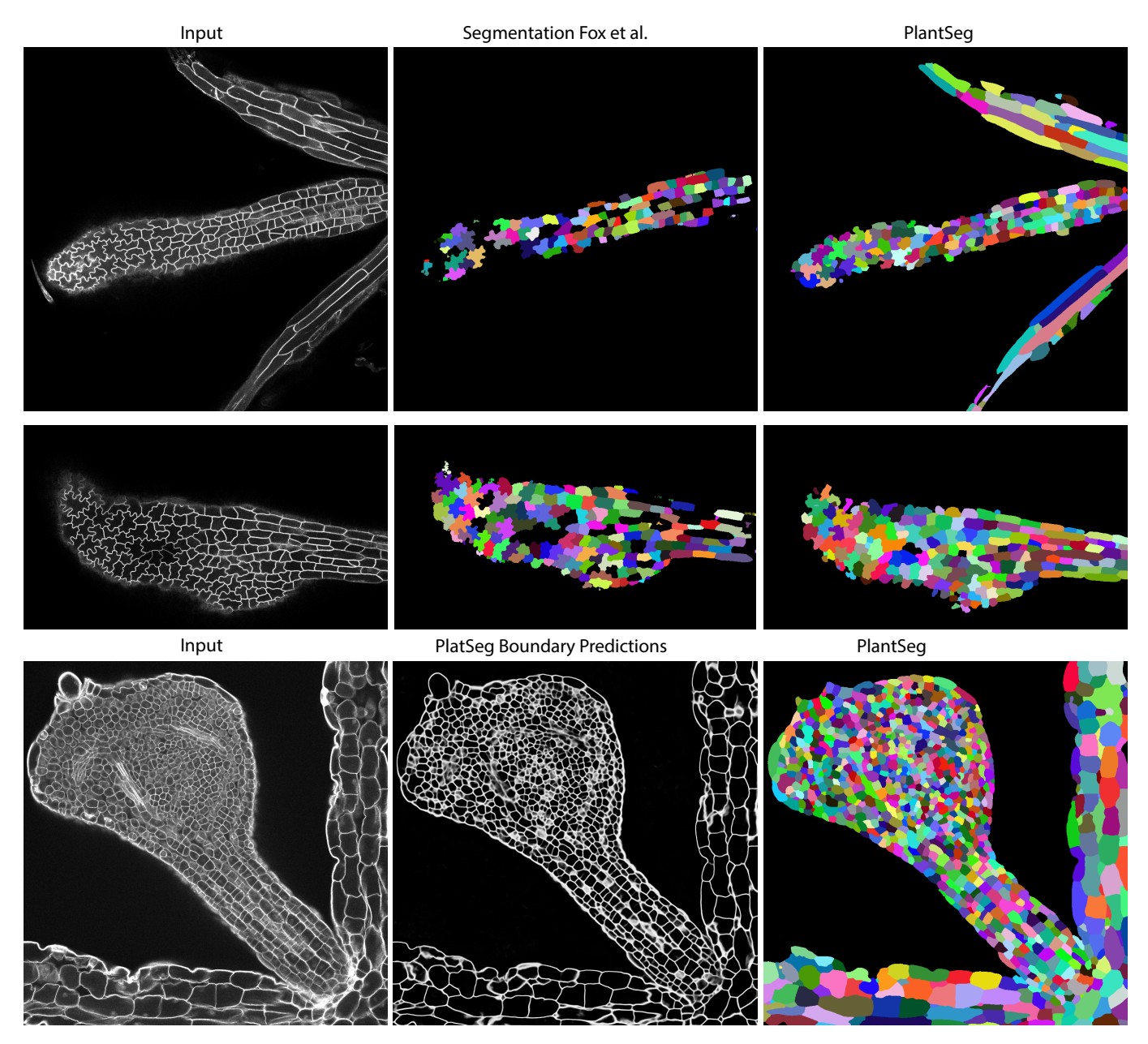

**Figure 5.** Qualitative results on the highly lobed epidermal cells from *Fox et al., 2018*. First two rows show the visual comparison between the SimpleITK (middle) and PlantSeg (right) segmentation on two different image stacks. PlantSeg's results on another sample is shown in the third row. In order to show pre-trained networks' ability to generalized to external data, we additionally depict PlantSeg's boundary predictions (third row, middle). We obtained the boundary predictions using the *generic-confocal-3d-unet* and segmented using GASP with default values. A value of 0.7 was chosen for the under/over segmentation factor.

## Asymmetric division of lateral root founder cells

*Arabidopsis thaliana* constantly forms lateral roots (LRs) that branch from the main root. These LRs arise from a handful of founder cells located in the pericycle, a layer of cells located deep within the primary root. Upon lateral root initiation, these founder cells invariably undergo a first asymmetric division giving rise to one small and one large daughter cell. Although the asymmetric nature of this division has long been reported (*Laskowski et al., 1995*; *Malamy and Benfey, 1997*) and its importance realised (*von Wangenheim et al., 2016*), it is still unknown how regular the volume

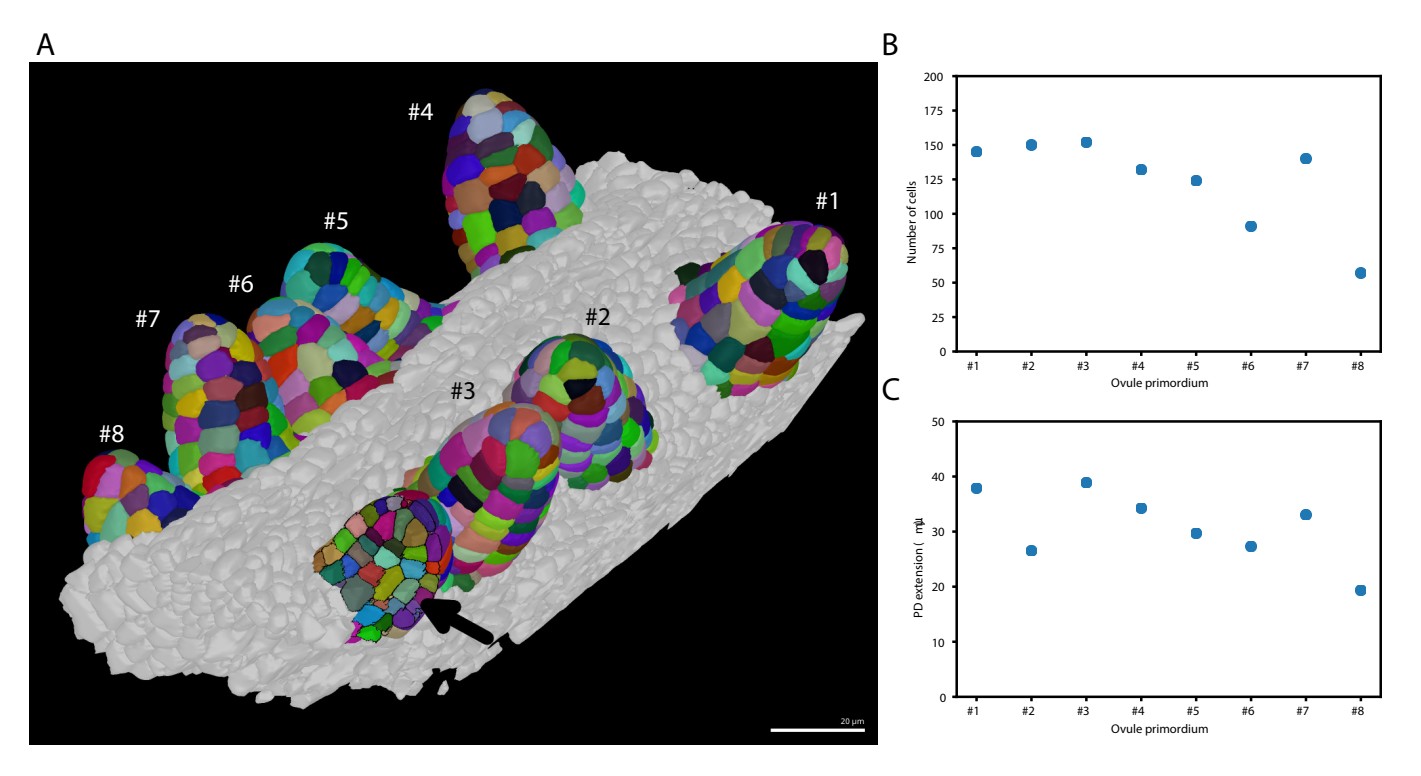

**Figure 6.** Ovule primordium formation in *Arabidopsis thaliana*. (**A**) 3D reconstructions of individually labeled stage 1 primordia of the same pistil are shown (stages according to *Schneitz et al., 1995*). The arrow indicates an optical mid-section through an unlabeled primordium revealing the internal cellular structure. The raw 3D image data were acquired by confocal laser scanning microscopy according to *Tofanelli et al., 2019*. Using MorphographX *Barbier de Reuille et al., 2015*, quantitative analysis was performed on the three-dimensional mesh obtained from the segmented image stack. Cells were manually labeled according to the ovule specimen (from #1 to #8). (**B, C**) Quantitative analysis of the ovule primordia shown in (**A**). (**B**) shows a graph depicting the total number of cells per primordium. (**C**) shows a graph depicting the proximal-distal (PD) extension of the individual primordia (distance from the base to the tip). Scale bar: 20 μm. Source files used for creation of the scatter plots (**B, C**) are available in the *Figure 6—source data 1*.

The online version of this article includes the following source data for figure 6:

**Source data 1.** Source data for panes B and C in *Figure 6*.

partitioning is between the daughter cells. We used the segmentation of the LR dataset produced by PlantSeg to quantify this ratio. The asymmetric divisions were identified by visual examination during the first 12 hr of the recording and the volumes of the two daughter cells retrieved (*Figure 7B*). The analysis of cell volume ratios confirmed that the first division of the founder cell is asymmetric with a volume ratio between the two daughter cells of $0.65 \pm 0.22$ (mean $\pm$ SD, $n = 23$) (*Figure 7C*).

## Epidermal cell volumes in a shoot apical meristem

Epidermal cell morphologies in the shoot apical meristem of *Arabidopsis thaliana* are genetically controlled and even subtle changes can have an impact on organogenesis and pattern formation. To quantify respective cell shapes and volumes in the newly identified *big cells in epidermis* (*bce*) mutant we used the PlantSeg package to analyze image volumes of six *Arabidopsis thaliana* meristems (three wild type and three *bce* specimens).

Inflorescence meristems of *Arabidopsis thaliana* were imaged using confocal las[er scanning microscopy after staining cell walls with DAPI. Image volumes $(167 \times 167 \times 45)$ were used to obtain 3D cell segmentations using PlantSeg: in this case a 3D UNet trained on the *Arabidopsis* ovules was used in combination with the Multicut algorithm. This segmentation procedure allows to determine epidermal cell volumes for wild-type (*Figure 8A*) and the *bce* mutant (*Figure 8B*). Cells within a radius of 33 μm around the manually selected center of the meristem (colored cells in *Figure 8A*

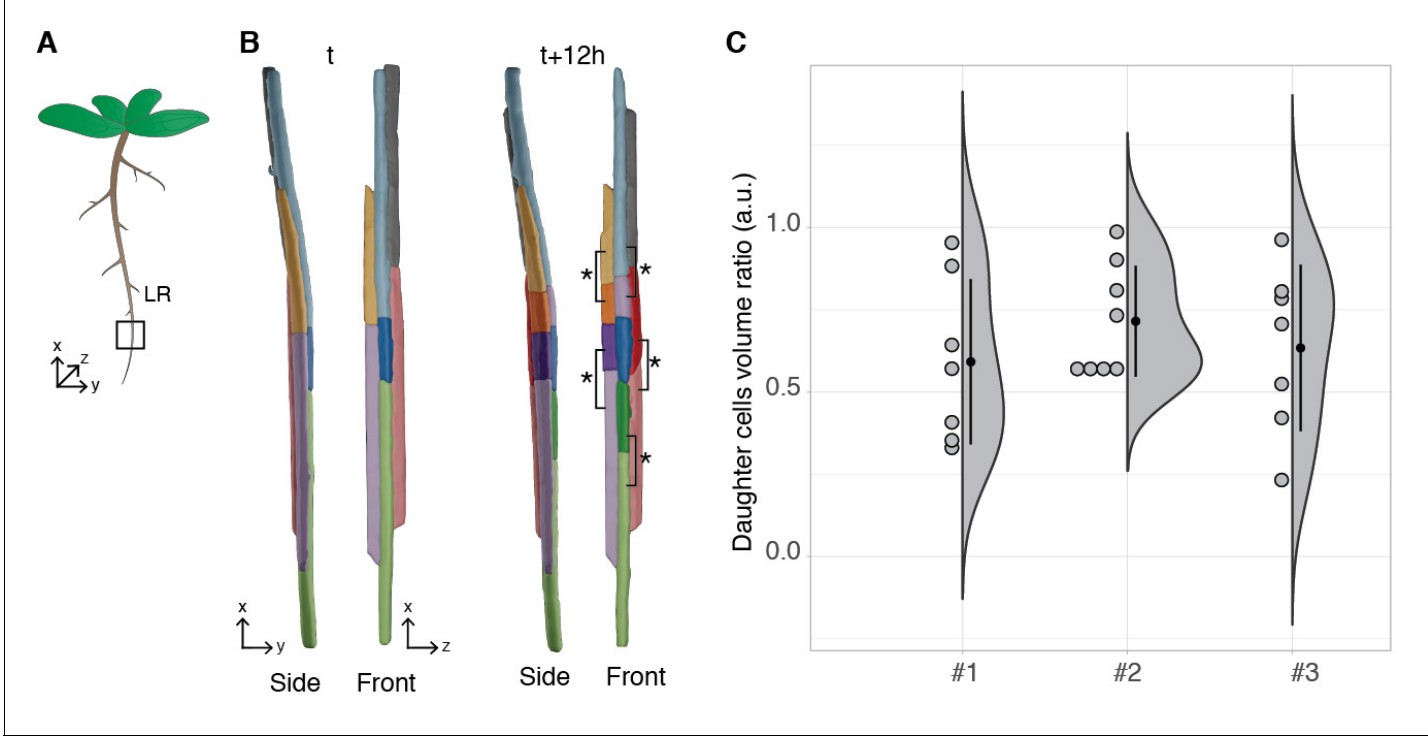

**Figure 7.** Asymmetric cell division of lateral root founder cells. (**A**) Schematic representation of *Arabidopsis thaliana* with lateral roots (LR). The box depicts the region of the main root that initiates LRs. (**B**) 3D reconstructions of LR founder cells seen from the side and from the front at the beginning of recording (**t**) and after 12 hr (*t+12*). The star and brackets indicate the two daughter cells resulting from the asymmetric division of a LR founder cell. (**C**) Half-violin plot of the distribution of the volume ratio between the daughter cells for three different movies (*#1*, *#2* and *#3*). The average ratio of 0.6 indicates that the cells divided asymmetrically. Source files used for analysis and violin plot creation are available in *Figure 7—source data 1*. The online version of this article includes the following source data for figure 7:

**Source data 1.** Source data for asymmetric cell division measurements in *Figure 7*.

*and B*) were used for the cell volume quantification shown in *Figure 8C*. The mean volume of epidermal cells in the *bce* mutant is increased by roughly 50% whereas overall meristem size is only slightly reduced which implicates changes in epidermal cell division in mutant meristems.

## Analysis of leaf growth and differentiation

Leaves are a useful system to study morphogenesis in the context of evolution because final organ forms of different species show considerable variation despite originating from almost indistinguishable buds (*Kierzkowski et al., 2019*). To track leaf growth, the same sample is imaged over the course of several days, covering a volume much larger than ovules or meristems. To reduce stress and growth arrest it is required to use relatively low resolution and laser intensity which makes an accurate full 3D segmentation more challenging. Because leaves grow mainly in two dimensions, their morphogenesis can be tracked on the organ surface. We therefore use the software platform MorphoGraphX which is specialized in creating and analyzing curved surface segmentations (*Barbier de Reuille et al., 2015*). It offers a semi-automatic surface segmentation pipeline using a seeded watershed algorithm (Figure 10A–C) but segmentation errors require extensive curation by the user (*Kierzkowski et al., 2019*). We tested whether PlantSeg can improve the segmentation pipeline of MorphoGraphX by using PlantSeg's membrane prediction and 3D segmented output files as additional input for creating the surface segmentation in MorphoGraphX. We used confocal laser scanning microscopy stacks of *Arabidopsis thaliana* and *Cardamine hirsuta* leaves fluorescently tagged at the cell boundaries (Figure 10A). Voxel sizes ranged from $0.33 \times 0.33 \times 0.5$ to $0.75 \times 0.75 \times 0.6 \mu m$.

We compared the auto-segmentation produced by MorphoGraphX using the original raw stacks as input (RawAutoSeg) to the one produced by MorphoGraphX using PlantSeg's wall prediction

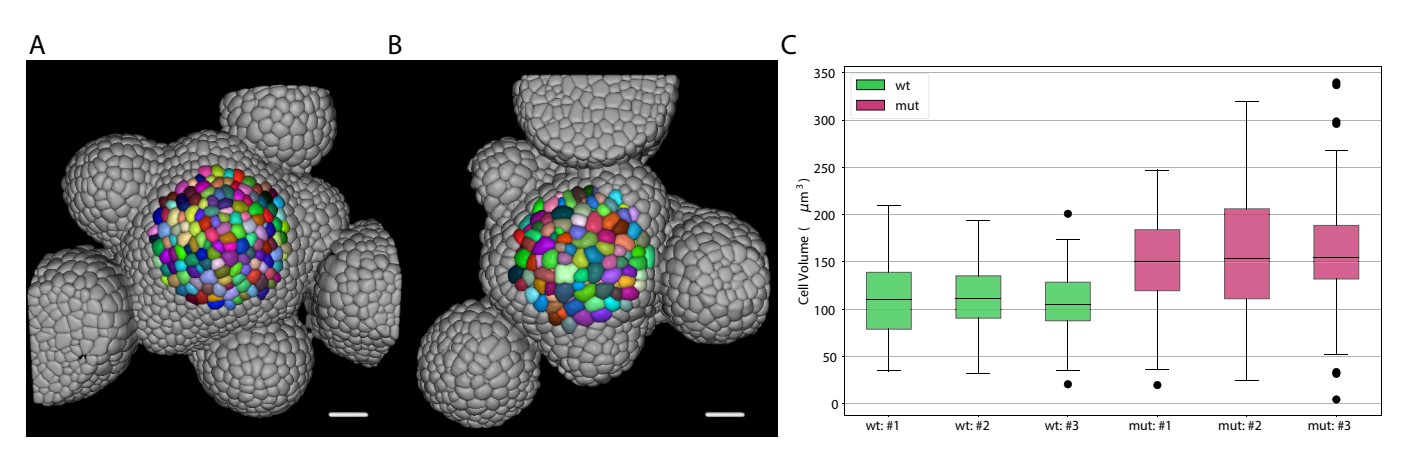

**Figure 8.** Volume of epidermal cell in the shoot apical meristem of Arabidopsis. Segmentation of epidermal cells in *wildtype* (A) and *bce* mutant (B). Cells located at the center of the meristem are colored. Scale bar: 20 μm. (C) Quantification of cell volume (μm$^3$) in three different *wildtype* and *bce* mutant specimens. Source files used for cell volume quantification are available in the ***Figure 8—source data 1***.

The online version of this article includes the following source data for figure 8:

**Source data 1.** Source data for volume measurements of epidermal cells in the shoot apical meristem (***Figure 8***).

(PredAutoSeg) or by projecting PlantSeg's 3D segmentation (Proj3D) (Figure 10B,C). We tested six different samples and computed the quality measures for the results of all different methods: ARand, $VOI_{merge}$ and $VOI_{split}$ as well as the accuracy (% of correctly segmented cells compared to the GT). The two methods using PlantSeg input produced lower ARand scores and higher accuracy than using the raw input (***Figure 9***). Therefore, combining PlantSeg with MorphographX produced segmentations more similar to the GT at the vertex and cell levels.

Next, we used the PlantSeg segmentation to measure growth over 24 hr at cellular resolution and compare differentiation in *A. thaliana* and *C. hirsuta* 600 μm-long leaves. Growth was slow in the midrib and distal margin cells, whereas the remaining blade displayed a gradient along the proximal-distal axis with the maximum values at the basal margin (***Figure 10D***). Tissue differentiation typically starts at the apex of leaves and progresses basipetally influencing this growth gradient. To compare this process between *A. thaliana* and *C. hirsuta* leaves, for each cell, we extracted its distance to the leaf base together with its area and lobeyness, attributes positively correlated with differentiation (***Kierzkowski et al., 2019***). Overall, *A. thaliana* leaves showed higher cell size and lobeyness, and this difference accentuated towards the apex, confirming earlier differentiation onset in this species (***Figure 10E,F***).

## Discussion

Taking advantage of the latest developments in machine learning and computer vision we created PlantSeg, a simple, powerful, and versatile tool for plant cell segmentation. Internally, it implements a two-step algorithm: the images are first passed through a state-of-the-art convolutional neural network to detect cell boundaries. In the second step, the detected boundaries are used to over-segment the image using the distance transform watershed and then a region adjacency graph of the image superpixels is constructed. The graph is partitioned to deliver accurate segmentation even for noisy live imaging of dense plant tissue.

PlantSeg was trained on confocal images of *Arabidopsis thaliana* ovules and light sheet images of the lateral root primordia and delivers high-quality segmentation on images from these datasets never seen during training as attested by both qualitative and quantitative benchmarks. We experimented with different U-Net designs and hyperparameters, as well as with different graph partitioning algorithms, to equip PlantSeg with the ones that generalize the best. This is illustrated by the excellent performance of PlantSeg without retraining of the CNNs on a variety of plant tissues and organs imaged using confocal microscopy (3D Cell Atlas Dataset) including the highly lobed

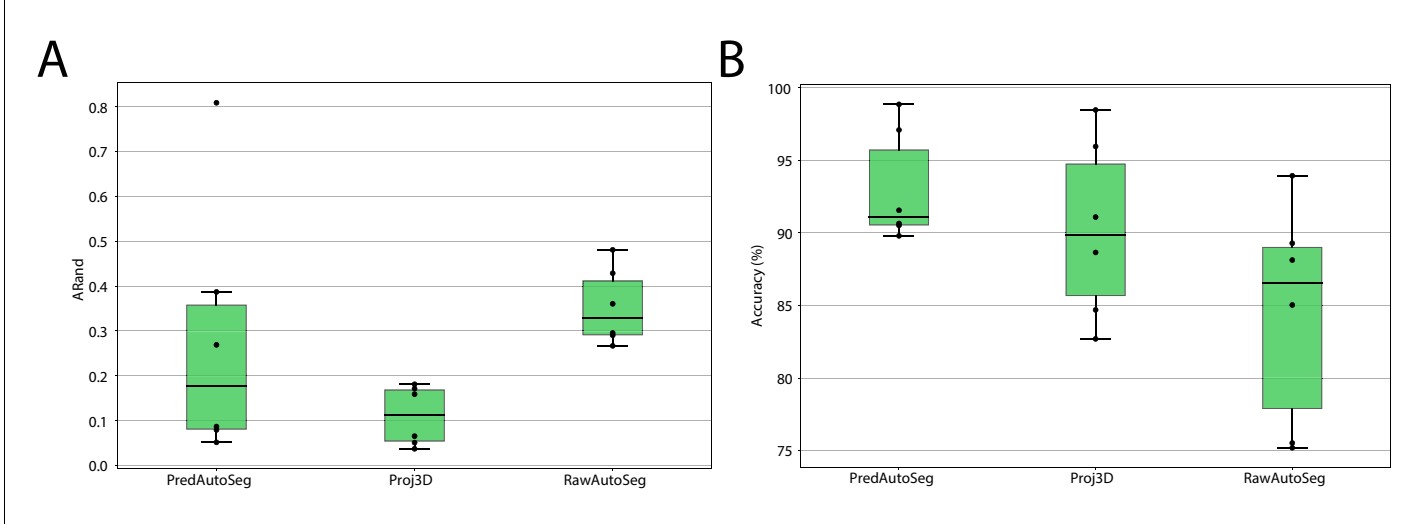

**Figure 9.** Leaf surface segmentation results. Reported are the ARand error (**A**) that assesses the overall segmentation quality and the accuracy (**B**) measured as percentage of correctly segmented cells (by manual assessment of a trained biologist). For more detailed results, see *Appendix 5—table 3*.

The online version of this article includes the following source data for figure 9:

**Source data 1.** Source data for leaf surface segmentation in *Figure 9*.

epidermal cells (*Fox et al., 2018*). This feature underlines the versatility of our approach for images presenting similar features to the ones used in training. In addition, PlantSeg comes with scripts to train a CNN on a new set of images and evaluate its performance. Given the importance of ground truth for training of CNNs we also provide instructions on how to generate ground truth in the Appendix 1: Groundtruth Creation. Besides the plant data, we compared PlantSeg to the state-of-the-art on an open benchmark for the segmentation of epithelial cells in the *Drosophila* wing disc. Using only the pre-trained networks, PlantSeg performance was shown to be close to the benchmark leaders, while additional training on challenge data has narrowed the gap even further.

We demonstrate the usefulness of PlantSeg on four concrete biological applications that require accurate extraction of cell geometries from complex, densely packed 3D tissues. First, PlantSeg allowed to sample the variability in the development of ovules in a given pistil and reveal that those develop in a relatively synchronous manner. Second, PlantSeg allowed the precise computation of the volumes of the daughter cells resulting from the asymmetric division of the lateral root founder cell. This division results in a large and a small daughter cells with volume ratio of $\sim \frac{2}{3}$ between them. Third, segmentation of the epidermal cells in the shoot apical meristem revealed that these cells are enlarged in the *bce* mutant compared to wild type. Finally, we showed that PlantSeg can be used to improve the automated surface segmentation of time-lapse leaf stacks which enables different downstream analyses such as growth tracking at cell resolution. Accurate and versatile extraction of cell outlines rendered possible by PlantSeg opens the door to rapid and robust quantitative morpho-metric analysis of plant cell geometry in complex tissues. This is particularly relevant given the central role plant cell shape plays in the control of cell growth and division (*Rasmussen and Bellinger, 2018*).

Unlike intensity-based segmentation methods used, for example, to extract DAPI-stained cell nuclei, our approach relies purely on boundary information derived from cell contour detection. While this approach grants access to the cell morphology and cell-cell interactions, it brings addi-tional challenges to the segmentation problem. Blurry or barely detectable boundaries lead to dis-continuities in the membrane structure predicted by the network, which in turn might cause cells to be under-segmented. The segmentation results produced by PlantSeg on new datasets are not fully perfect and still require proof-reading to reach 100% accuracy. For our experiments we used Pain-tera (*Hanslovsky et al., 2019*) for manually correcting the 3D segmentation results. Importantly the newly proof-read results can then be used to train a better network that can be applied to this type

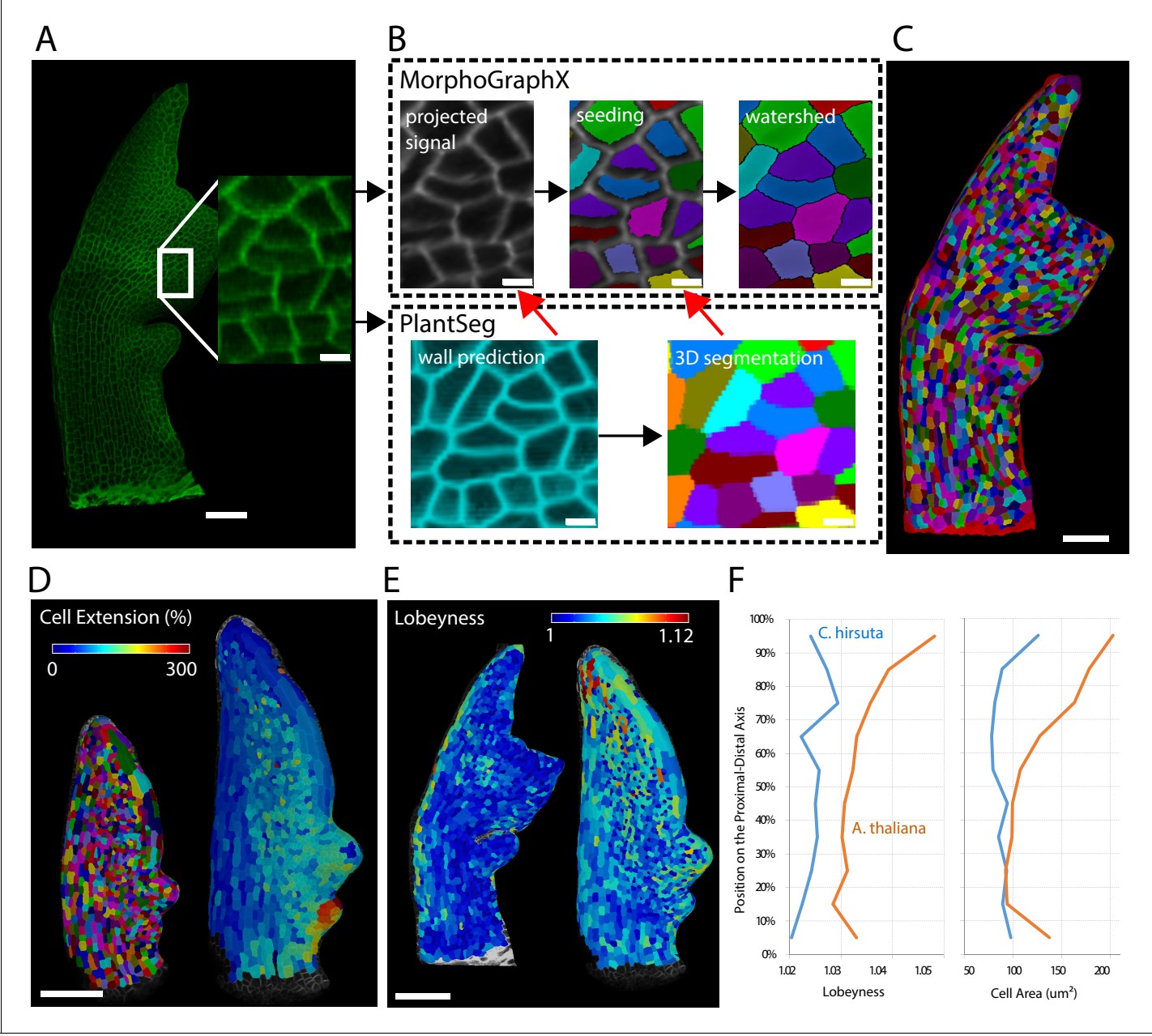

**Figure 10.** Creation of cellular segmentations of leaf surfaces and downstream quantitative analyses. (**A–C**) Generation of a surface segmentation of a *C. hirsuta* leaf in MorphoGraphX assisted by PlantSeg. (**A**) Confocal image of a 5-day-old *C. hirsuta* leaf (leaf number 5) with an enlarged region. (**B**) Top: Segmentation pipeline of MorphoGraphX: a surface mesh is extracted from the raw confocal data and used as a canvas to project the epidermis signal. A seed is placed in each cell on the surface for watershed segmentation. Bottom: PlantSeg facilitates the segmentation process in two different ways (red arrows): By creating clean wall signals which can be projected onto the mesh instead of the noisy raw data and by projecting the labels of the 3D segmentation onto the surface to obtain accurate seeds for the cells. Both methods reduce segmentation errors with the first method to do so more efficiently. (**C**) Fully segmented mesh in MorphoGraphX. (**D–F**) Quantification of cell parameters from segmented meshes. (**D**) Heatmap of cell growth in an *A. thaliana* 8th-leaf 4 to 5 days after emergence. (**E**) Comparison of cell lobeyness between *A. thaliana* and *C. hirsuta* 600 μm-long leaves. (**F**) Average cell lobeyness and area in *A. thaliana* and *C. hirsuta* binned by cell position along the leaf proximal-distal axis. Scale bars: 50 μm (**A, C**), 100 μm (**D, E**), 5 μm (inset in A, (**B**). Source files used for generating quantitative results (**D–F**) are available in *Figure 10—source data 1*. The online version of this article includes the following source data for figure 10:

**Source data 1.** Source data for pane F in *Figure 10* (cell area and lobeyness analysis).

of data in the future (see the Appendix 1: Groundtruth Creation for an overview of this process). If nuclei are imaged along with cell contours, nuclear signal can be leveraged for improving the segmentation as we have explored in *Pape et al., 2019* (see Appendix 2: Exploiting nuclei staining to improve the lateral root cells' segmentation for detailed procedure). In future work, we envision developing new semi-supervised approaches that would exploit the vast amounts of unlabeled data available in the plant imaging community.

During the development of PlantSeg, we realised that very few benchmark datasets were available to the community for plant cell segmentation tasks, a notable exception being the 3D Tissue Atlas (*Bassel, 2019*). To address this gap, we publicly release our sets of images and the corresponding hand-curated ground truth in the hope to catalyse future development and evaluation of cell instance segmentation algorithms.

## Materials and methods

### Biological material and imaging

Imaging of the *Arabidopsis thaliana* ovules was performed as described in *Tofanelli et al., 2019*. Imaging of the shoot apical meristem was performed as previously described *von Wangenheim et al., 2014*; *Heisler and Ohno, 2014* with a confocal laser scanning microscope (Nikon A1, $25 \times$ NA = 1.1) after staining cell walls with DAPI (0.2 mg/ml).

For imaging of *Arabidopsis thaliana* lateral root, seedlings of the line sC111 (*UB10$_{pro}$ :: PIP 1,4-3 $\times$ GFP/GAT A23$_{pro}$ :: H2B : 3 $\times$ mCherry/DR5v2$_{pro}$ :: 3 $\times$ YFPnls/RPS5A$_{pro}$ :: dtTomato : NLS*, described in *Vilches Barro et al., 2019*) were used at 5 day post germination. Sterilized seeds were germinated on top of 4.5 mm long Blaubrand micropipettes (Cat 708744; 100 μl) that were immobilised on the bottom of a petri dish and covered with $\frac{1}{2}$ MS-phytagel (*Maizel et al., 2011*). Before sowing, the top of the micropipettes is exposed by removing the phytagel with a razor blade and one seed is sowed per micropipette. Plates were stratified for two days and transferred to a growth incubator (23°C, 16 h day light). Imaging was performed on a Luxendo MuViSPIM (https://luxendo. eu/products/muvi-spim/) equipped with two 10×NA=0.3for illumination and 40×NA=0.8for detection. The following settings were used for imaging: image size $2048 \times 2048$, exposure time 75 ms, channel #1 illumination 488 nm 10% power, detection 497–553 nm band pass filter, channel #2 illumination 561 nm 10% power, detection 610–628 nm band pass filter. Stacks encompassing the whole volume of the root were acquired every 30 min. Images from the two cameras were fused using the Luxendo Image processing tool and registered to correct any 3D drift using the BigDataProcessor (*Tischer et al., 2019*) in Fiji (*Schindelin et al., 2012*).

Leaves were grown and imaged as described previously (*Kierzkowski et al., 2019*). Cells were visualized either by expressing of UBQ10::acyl:YFP (*Willis et al., 2016*) or by staining with 10 mg/mL propidium iodide for 15 min. The bce mutant is a yet uncharacterised recessive mutant obtained in J. Lohmanns lab. The phenotype was observed after T-DNA transformation of Arabidopsis Col-0 plants.

### Creation of leaf surface segmentations

To compare the segmentations created by MorphoGraphX alone with the ones using PlantSeg's files as input, we first obtained a ground-truth segmentation using the MorphographX auto-segmentation pipeline as described in *Strauss et al., 2019* (*Figure 10B*) and manually fixed all segmentation errors using processes in MorphoGraphX. We then fed the confocal stacks to PlantSeg to compute wall predictions and 3D segmentations using the network trained on the ovule confocal data and the GASP method. Note that for samples with weaker cell wall signal we processed the raw input data in MorphoGraphX by adding a 2 μm thick layer of signal under the surface mesh and fed these to PlantSeg which tended to improve the PlantSeg output greatly. We then created surface segmentation using three methods: First, using the raw stack and the auto-segmentation pipeline in MorphoGraphX (method RawAutoSeg, *Figure 10B*, top). Second, using PlantSeg's wall prediction values as input for the auto-segmentation process in MorphoGraphX (method PredAutoSeg, *Figure 10B*, left red arrow) and third, using PlantSeg's fully segmented stack and projecting the resulting 3D labels onto the surface mesh using a custom process in MorphoGraphX (method Proj3D, *Figure 10B*, right red arrow).

## Neural network training and inference

### Training

2D and 3D U-Nets were trained to predict the binary mask of cell boundaries. Ground truth cell contours where obtained by taking the ground truth label volume, finding a two voxels-thick boundaries between labeled regions (*find_boundaries*(·) function from the Scikit-image package (*van der Walt et al., 2014*) and applying a Gaussian blur on the resulting boundary image. Gaussian smoothing reduces the high frequency components in the boundary image, which helps prevent over-fitting and makes the boundaries thicker, increasing the amount of foreground signal during training. Transforming the label image $\mathcal{S}_\mathbf{x}$ into the boundary image $\mathcal{I}_\mathbf{x}$ is depicted in *Equation 1*.

$$\mathcal{I}_\mathbf{x} = \begin{cases} 1 & \text{if } \Phi(\mathcal{S}_\mathbf{x}) * G_\sigma > 0.5 \\ 0 & \text{otherwise} \end{cases} \tag{1}$$

Where $\Phi(\cdot)$ transforms the labeled volume into the boundary image, $G_\sigma$ is the isotropic Gaussian kernel and * denotes a convolution operator. We use $\sigma = 1.0$ in our experiments. Both standard and residual U-Net architectures were trained using Adam optimizer (*Kingma and Ba, 2014*) with $\beta_1 = 0.9, \beta_2 = 0.999$, L2 penalty of 0.00001 and initial learning rate $\epsilon = 0.0002$. Networks were trained until convergence for 150K iterations, using the PyTorch framework (*Paszke et al., 2019*) on 8 NVIDIA GeForce RTX 2080 Ti GPUs. For validation during training, we used the adjusted Rand error computed between the ground truth segmentation and segmentation obtained by thresholding the probability maps predicted by the network and running the connected components algorithm. The learning rate was being reduced by a factor of 2 once the learning stagnated during training, that is, no improvements were observed on the validation set for a given number of iterations. We choose as best network the one with lowest Arand error values. For training with small patch sizes we used 4 patches of shape $100 \times 100 \times 80$ and batch normalization (*Ioffe and Szegedy, 2015*) per network iteration. When training with a single large patch (size $170 \times 170 \times 80$), we used group normalization layers (*Wu and He, 2018*) instead of batch normalization. The reason is that batch normalization with a single patch per iteration becomes an instance normalization (*Ulyanov et al., 2016*) and makes the estimated batch statistics weaker. All networks use the same layer ordering where the normalization layer is followed by the 3D convolution and a rectified linear unit (ReLU) activation. This order of layers consistently performed better than alternative orderings. During training and inference, input images were standardized by subtracting mean intensity and dividing by the standard deviation. We used random horizontal and vertical flips, random rotations in the XY-plane, elastic deformations (*Ronneberger et al., 2015*) and noise augmentations (additive Gaussian, additive Poisson) of the input image during training in order to increase network generalization on unseen data. The performance of CNNs is sensitive to changes in voxel size and object sizes between training and test images (*van Noord and Postma, 2017*). We thus also trained the networks using the original datasets downscaled by a factor of 2 and 3 in the XY dimension.

3D U-Nets trained at different scales of our two core datasets (light-sheet lateral root, confocal ovules) are made available as part of the PlantSeg package. All released networks were trained according to the procedure described above using a combination of binary cross-entropy and Dice loss:

$$\mathcal{L} = \alpha \mathcal{L}_{BCE} + \beta \mathcal{L}_{Dice} \tag{2}$$

(we set $\alpha = 1$, $\beta = 1$ in our experiments) and follow the standard U-Net architecture (*Ronneberger et al., 2015*) with two minor modifications: batch normalization (*Ioffe and Szegedy, 2015*) is replaced by group normalization (*Wu and He, 2018*) and same convolutions are used instead of valid convolutions. For completeness we also publish 2D U-Nets trained using the Z-slices from the original 3D stacks, enabling segmentation of 2D images with PlantSeg.

### Inference

During inference we used mirror padding on the input image to improve the prediction at the boundaries of the volume. We kept the same patch sizes as during training since increasing it during inference might lead to lower quality of the predictions, especially on the borders of the patch. We also parse the volume patch-by-patch with a 50% overlap between consecutive tiles and average the

probability maps. This strategy prevents checkerboard artifacts and reduces noise in the final prediction.

The code used for training and inference can be found at *Wolny, 2020b* https://github.com/wolny/pytorch-3dunet copy archived at https://github.com/elifesciences-publications/pytorch-3dunet.

## Segmentation using graph partitioning

The boundary predictions produced by the CNN are treated as a graph $G(V, E)$, where nodes *V* are represented by the image voxels, and the edges *E* connect adjacent voxels. The weight $w \in R^+$ of each edge is derived from the boundary probability maps. On this graph we first performed an over-segmentation by running the DT watershed (*Roerdink and Meijster, 2000*). For this, we threshold the boundary probability maps at a given value δ to get a binary image (δ = 0.4 was chosen empirically in our experiments). Then we compute the distance transform from the binary boundary image, apply a Gaussian smoothing (*sigma* = 2.0) and assign a seed to every local minimum in the resulting distance transform map. Finally we remove small regions (<50 voxels). Standalone DT watershed already delivers accurate segmentation and can be used as is in simple cases when, for example noise is low and/or boundaries are sharp.

For Multicut (*Kappes et al., 2011*), GASP (*Bailoni et al., 2019*), and Mutex watershed (*Wolf et al., 2018*) algorithms, we used the DT watershed as an input. Although all three algorithms could be run directly on the boundary predictions produced by the CNN (voxel level), we choose to run them on a region adjacency graph (RAG) derived from the DT watershed to reduce the computation time. In the region adjacency graph each node represents a region and edges connect adjacent regions. We compute edge weights by using the mean value of the probabilities maps along the boundary. We then run Multicut, GASP or Mutex watershed with a hyperparameter *beta* = 0.6 that balances over- and under-segmentation (with higher β tending to over-segment). As a general guideline for choosing the partitioning strategy on a new data is to start with GASP algorithm, which is the most generic. If needed, one may try to improve the results with multicut or mutex watershed. If none of the three strategies give satisfactory segmentation results we recommend to over-segment provided stack using the distance transform watershed and proofread the result manually using Paintera software (*Hanslovsky et al., 2019*).

A detailed overview of the parameters exposed via the PlantSeg's UI can be found on the project's GitHub page https://github.com/hci-unihd/plant-seg as well as in *Appendix 6—table 3*.

## Metrics used for evaluation

For the boundary predictions we used precision (number of pixels positively predicted as boundary divided by the number of boundary pixels in the ground truth), recall (number of positively predicted boundary pixels divided by the sum of positively and negatively predicted boundary pixels) and F1 score

$$F1 = 2 \cdot \frac{Precision \cdot Recall}{Precision + Recall}. \tag{3}$$

For the final segmentation, we used the inverse of the Adjusted Rand Index (AdjRand) *Rand, 1971* defined as $\mathrm{ARanderror} = 1 - \mathrm{AdjRand}$ (*CREMI, 2017*) which measures the distance between two clustering as global measure of accuracy between PlantSeg prediction and ground truth. An ARand error of 0 means that the PlantSeg results are identical to the ground truth, whereas one shows no correlation between the ground truth and the segmentation results. To quantify the rate of merge and split errors, we used the Variation of Information (VOI) which is an entropy based measure of clustering quality (*Meila, 2005*). It is defined as:

$$\mathrm{VOI} = H(\mathrm{seg}|\mathrm{GT}) + H(\mathrm{GT}|\mathrm{seg}), \tag{4}$$

where *H* is the conditional entropy function and the Seg and GT the predicted segmentation and ground truth segmentation respectively. $H(\mathrm{seg}|\mathrm{GT})$ defines the split mistakes (*VOI_{split}*) whereas $H(\mathrm{GT}|\mathrm{Seg})$ corresponds to the merge mistakes (*VOI_{merge}*).

## Acknowledgements

We thank Kerem Celikay, Melanie Holzapfel and Boyko Vodenicharski for their help in the early steps of the project. We are grateful to Natalie Dye from MPI-CBG for sharing the flywing data and annotations. We further acknowledge the support of the Center for Advanced Light Microscopy (CALM) at the TUM School of Life Sciences. GWB and SD-N were funded by Leverhulme Grant RPG-2016–049. CP and AK were funded by Baden-Wuerttemberg Stiftung. MT acknowledges support from the German Federal Ministry of Education and Research (BMBF, grant number 031B0189B), in the context of the project 'Enhancing Crop Photosynthesis (EnCroPho)". This work was supported by the DFG FOR2581 to the Hamprecht (P3), Kreshuk (P3), Lohmann (P5), Maizel (P6), Schneitz (P7) and Tsiantis (P9) labs.

## Additional information

### Funding

| Funder | Grant reference number | Author |
|---|---|---|
| Deutsche Forschungsgemeinschaft | FOR2581 | Jan U Lohmann<br>Miltos Tsiantis<br>Fred A Hamprecht<br>Kay Schneitz<br>Alexis Maizel<br>Anna Kreshuk |
| Leverhulme Trust | RPG-2016-049 | George Bassel |

The funders had no role in study design, data collection and interpretation, or the decision to submit the work for publication.

### Author contributions

Adrian Wolny, Conceptualization, Data curation, Software, Formal analysis, Supervision, Funding acquisition, Validation, Investigation, Visualization, Methodology, Writing - original draft, Project administration, Writing - review and editing; Lorenzo Cerrone, Athul Vijayan, Conceptualization, Data curation, Software, Formal analysis, Validation, Investigation, Visualization, Methodology, Writing - original draft, Writing - review and editing; Rachele Tofanelli, Amaya Vilches Barro, Data curation, Formal analysis; Marion Louveaux, Rena Lymbouridou, Susanne S Steigleder, Data curation; Christian Wenzl, Data curation, Visualization, Writing - original draft; Sören Strauss, David Wilson-Sánchez, Data curation, Software, Formal analysis, Investigation, Visualization, Writing - original draft; Constantin Pape, Data curation, Software, Methodology; Alberto Bailoni, Software, Methodology; Salva Duran-Nebreda, Resources, Software; George W Bassel, Resources; Jan U Lohmann, Resources, Supervision; Miltos Tsiantis, Conceptualization, Resources, Supervision; Fred A Hamprecht, Conceptualization, Resources, Software, Supervision, Methodology, Writing - original draft; Kay Schneitz, Conceptualization, Software, Formal analysis, Supervision, Methodology, Writing - original draft; Alexis Maizel, Conceptualization, Software, Formal analysis, Supervision, Funding acquisition, Investigation, Methodology, Writing - original draft, Project administration; Anna Kreshuk, Conceptualization, Software, Formal analysis, Supervision, Funding acquisition, Validation, Investigation, Methodology, Writing - original draft, Project administration

### Author ORCIDs

Adrian Wolny https://orcid.org/0000-0003-2794-4266
Rachele Tofanelli http://orcid.org/0000-0002-5196-1122
Jan U Lohmann https://orcid.org/0000-0003-3667-187X
Kay Schneitz https://orcid.org/0000-0001-6688-0539
Anna Kreshuk https://orcid.org/0000-0003-1334-6388

### Decision letter and Author response

Decision letter https://doi.org/10.7554/eLife.57613.sa1
Author response https://doi.org/10.7554/eLife.57613.sa2

# Additional files

## Supplementary files
• Transparent reporting form

## Data availability

All data used in this study have been deposited in Open Science Framework: https://osf.io/uzq3w.

The following datasets were generated:

| Author(s) | Year | Dataset title | Dataset URL | Database and Identifier |
|---|---|---|---|---|
| Wilson-Sánchez D, Lymbouridou R, Strauss S, Tsiantis M | 2019 | CLSM Leaf | https://osf.io/kfx3d/ | Open Science Framework, 10.17605/OSF.IO/KFX3D |
| Wenzl C, Lohmann JU | 2019 | Inflorescence Meristem | https://osf.io/295su/ | Open Science Framework, 10.17605/OSF.IO/295SU |
| Louveaux M, Maizel A | 2019 | *A. Thaliana* Lateral Root | https://osf.io/2rszy/ | Open Science Framework, 10.17605/OSF.IO/2RSZY |
| Tofanelli R, Vijayan A, Schneitz K | 2019 | *A. Thaliana* Ovules | https://osf.io/w38uf/ | Open Science Framework, 10.17605/OSF.IO/W38UF |

The following previously published dataset was used:

| Author(s) | Year | Dataset title | Dataset URL | Database and Identifier |
|---|---|---|---|---|
| Duran-Nebreda S, Bassel G | 2019 | Arabidopsis 3D Digital Tissue Atlas | https://osf.io/fzr56/ | Open Science Framework, OSF |

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

## Appendix 1

### Groundtruth creation

Training state-of-the-art deep neural network for semantic segmentation requires a large amount of densely annotated samples. Groundtruth creation for the ovule dataset has been described previously (*Tofanelli et al., 2019*), we briefly describe here how the dense groundtruth labeling of cells for the lateral root was generated.

We bootstrapped the process using the Autocontext Workflow (*Tu and Bai, 2010*) of the opensource ilastik software (*Berg et al., 2019*) which is used to segment cell boundaries from sparse user input (scribbles). It is followed by the ilastik's multicut workflow (*Beier et al., 2017*) which takes the boundary segmentation image and produces the cell instance segmentation. These initial segmentation results were iteratively refined (see *Appendix 1—figure 1*). First, the segmentation is manually proofread in a few selected regions of interest using the open-source Paintera software (*Hanslovsky et al., 2019*). Second, a state-of-the-art neural network is trained for boundary detection on the manually corrected regions. Third, PlantSeg framework consisting of neural network prediction and image partitioning algorithm is applied to the entire dataset resulting in a more refined segmentation. The 3-step iterative process was repeated until an instance segmentation of satisfactory quality was reached. A final round of manual proofreading with Paintera is performed to finalize the groundtruth.

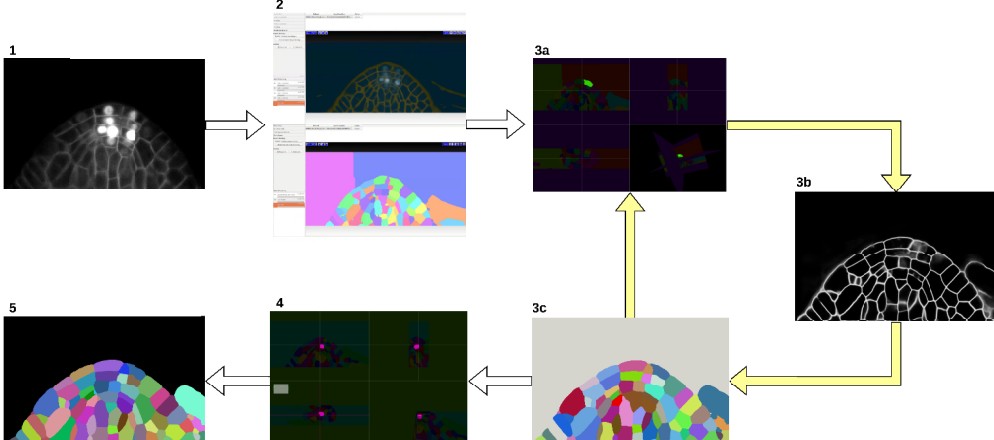

**Appendix 1—figure 1.** Groundtruth creation process. Starting from the input image (1), an initial segmentation is obtained using ilastik Autocontext followed by the ilastik multicut workflow (2). Paintera is used to proofread the segmentation (3a) which is used for training a 3D UNet for boundary detection (3b). A graph partitioning algorithm is used to segment the volume (3 c). Steps 3a, 3b and 3 c are iterated until a final round of proofreading with Paintera (4) and the generation of satisfactory final groundtruth labels (5).

## Appendix 2

### Exploiting nuclei staining to improve the lateral root cells' segmentation

The lateral root dataset contains a nuclei marker in a separate channel. In such cases we can take advantage of the fact that a cell contains only one nucleus and use this information as an additional clue during segmentation.

For this, we first segmented the nuclei using a simple and accurate segmentation pipeline that consists of a 3D U-Net trained to predict the binary nuclei mask followed by thresholding of the probability maps and connected components. We then incorporated this additional information into the multicut formulation, called lifted multicut (*Pape et al., 2019*; *Horňáková et al., 2017*), where additional repulsive edges are introduced between the nodes in the graph corresponding to the different nuclei segmented from the second channel.

We compared the scores of this lifted multicut algorithm to the scores for GASP, multicut and mutex watershed (see *Appendix 5—table 2*). We see that lifted multicut outperforms not only the standard multicut, but also all the other algorithms. This is because lifted multicut is able to separate two cells incorrectly merged into one region by the segmentation algorithm, as long as the region contains the two different nuclei instances corresponding to the merged cells.

A 3D U-Net trained to predict nuclei mask is available in the PlantSeg package. Lifted multicut segmentation can be executed via the PlantSeg's command line interface. We refer to the project's GitHub page for instructions.

## Appendix 3

### Performance of PlantSeg on an independent reference benchmark

To test the versatility of our approach, we assessed PlantSeg performance on a non-plant dataset consisting of 2D+t videos of membrane-stained developing *Drosophila* epithelial cells (*Aigouy et al., 2016*). The benchmark dataset and the results of four state-of-the-art segmentation pipelines are reported in *Funke et al., 2019b*. Treating the movie sequence as 3D volumetric images not only resembles the plant cell images shown in our study, but also allows to pose the 2D+t segmentation as a standard 3D segmentation problem.

We compared the performance of PlantSeg on the 8 movies of this dataset to the four reported pipelines: MALA (*Funke et al., 2017*), Flood Filling Networks (FFN) (*Januszewski et al., 2016*), Moral Lineage Tracing (MLT) (*Jug et al., 2015*; *Rempfler et al., 2017*) and Tissue Analyzer (TA) (*Funke et al., 2019b*). On our side, we evaluate two PlantSeg predictions: for the first one, we use the boundary detection network trained on the *Arabidopsis thaliana* ovules. This experiment gives an estimate of how well our pre-trained networks generalize to non-plant tissues. For the second evaluation, we retrain the network on the training data of the benchmark and obtain an estimate of the overall PlantSeg approach accuracy on non-plant data. Note that unlike other methods reported in the benchmark, we do not introduce any changes to account for the data being 2D+t rather than 3D, that is, we do not enforce the lineages to be moral as the authors of *Funke et al., 2019b* did with their segmentation methods.

For the first experiment, peripodial cells were segmented using the 3D U-Net trained on the ovule dataset together with GASP segmentation, whereas proper disc cells were segmented with 2D U-Net trained on the ovule dataset in combination with Multicut algorithm. Both networks are part of the PlantSeg package. Qualitative results of our pipeline are shown in *Figure 1*: PlantSeg produces very good segmentations on both the peripodial and proper imaginal disc cells. A few over-segmentation (peripodial cells) and under-segmentation (proper disc) errors are marked in the figure. This impression is confirmed by quantitative benchmark results in *Appendix 3—table 1*.

For the second experiment, we trained the network on the ground truth labels included in the benchmark (*PlantSeg (trained)*). Here, our pipeline is comparable to state-of-the-art. The difference in SEG metric between 'vanilla' PlantSeg and *PlantSeg (trained)* is 6.9 percent points on average, which suggests that for datasets sufficiently different from the ones PlantSeg networks were trained on, re-training the models might be necessary. Looking at the average run-times of the methods reported in the benchmark shows that PlantSeg pipeline is clearly the fastest approach with the average run-time of 3 min per movie when run on a server with a modern GPU versus 35 min (MALA), 42 min (MLT) and 90 min (FFN).

Thus, PlantSeg achieves results which are competitive with other proven methods in terms of accuracy, without explicitly training the boundary detection networks on the epithelial cell ground truth or accounting for the 2d+t nature of the data. PlantSeg outperforms all methods in term of computing time.

**Appendix 3—table 1.** Epithelial Cell Benchmark results.
We compare PlantSeg to four other methods using the standard SEG metric (*Maška et al., 2014*) calculated as the mean of the Jaccard indices between the reference and the segmented cells in a given movie (higher is better). Mean and standard deviation of the SEG score are reported for peripodial (three movies) and proper disc (five movies) cells. Additionally we report the scores of PlantSeg pipeline executed with a network trained explicitly on the epithelial cell dataset (last row).

| Method | Peripodial | Proper disc |
|---|---|---|
| MALA | 0.907 0.029 | 0.817 0.009 |
| FFN | 0.879 0.035 | 0.796 0.013 |
| MLT-GLA | 0.904 0.026 | 0.818 0.010 |
| TA | - | 0.758 0.009 |
| PlantSeg | 0.787 0.063 | 0.761 0.035 |
| PlantSeg (trained) | 0.885 0.056 | 0.800 0.015 |

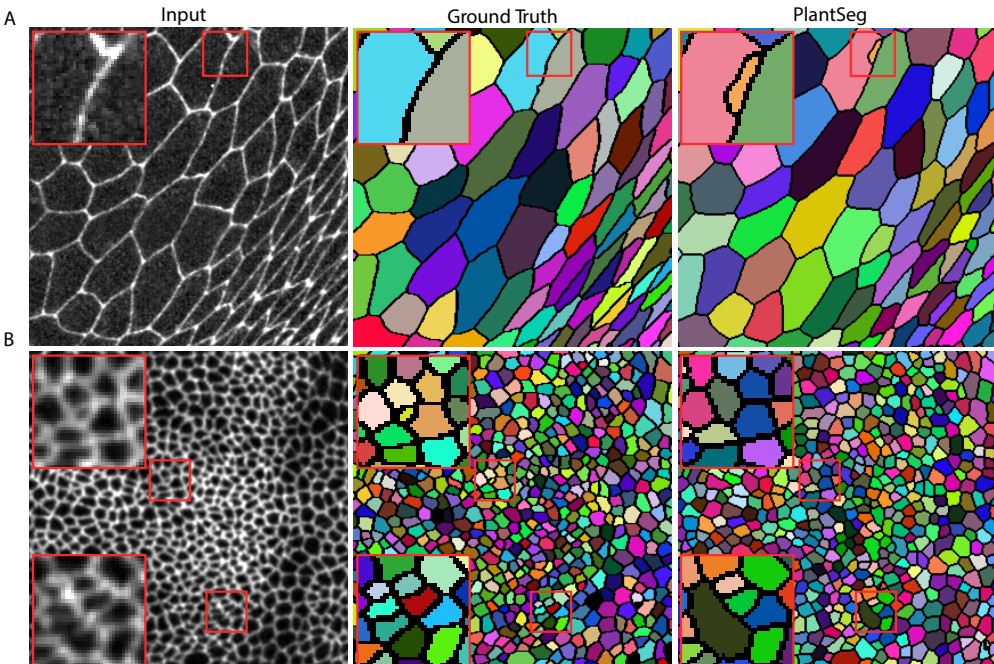

**Appendix 3—figure 1.** Qualitative results on the Epithelial Cell Benchmark. From top to bottom: Peripodial cells (**A**), Proper disc cells (**B**). From left to right: raw data, groundtruth segmentation, PlantSeg segmentation results. PlantSeg provides accurate segmentation of both tissue types using only the networks pre-trained on the *Arabidopsis* ovules dataset. Red rectangles show sample over-segmentation (**A**) and under-segmentation (**B**) errors. Boundaries between segmented regions are introduced for clarity and they are not present in the pipeline output.

# Appendix 4

## Supplemental figures

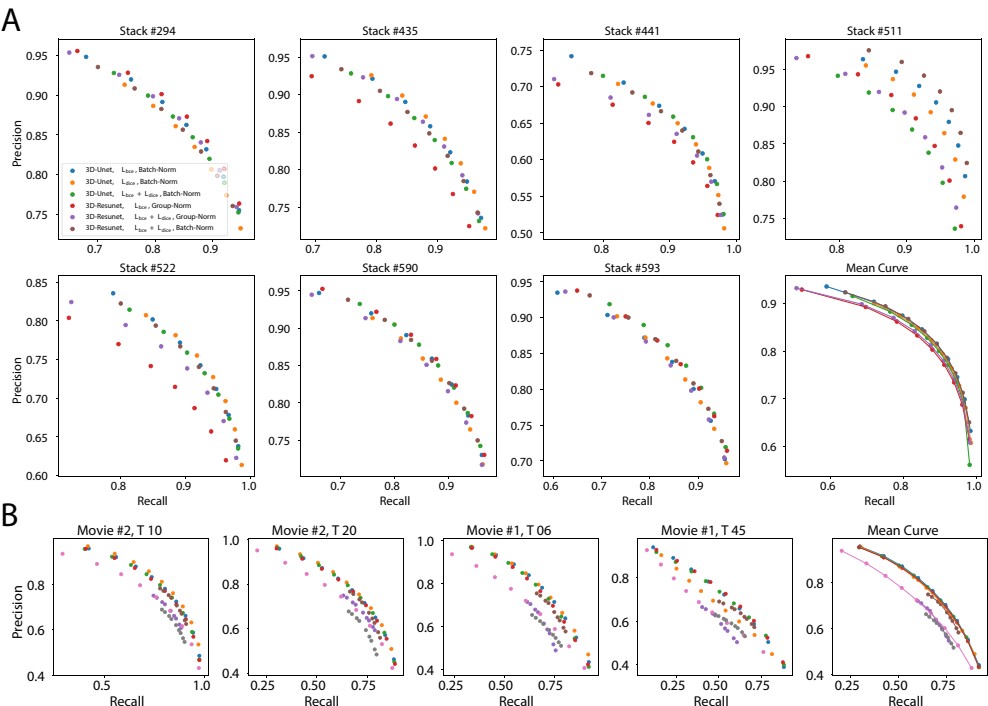

**Appendix 4—figure 1.** Precision-recall curves on individual stacks for different CNN variants on the ovule (A) and lateral root primordia (B) datasets. Efficiency of boundary prediction was assessed for seven training procedures that sample different type of architecture (3D U-Net *vs.* 3D Residual U-Net), loss function (BCE *vs.* Dice *vs.* BCE-Dice)) and normalization (Group-Norm (GN) *vs.* Batch-Norm (BN)). The larger the area under the curve, the better the precision. Source files used to generate the precision-recall curves are available in the *Appendix 4—figure 1—source data 1*.

The online version of this article includes the following source data is available for figure 1:

**Appendix 4—figure 1—source data 1.** Source data for precision/recall curves of different CNN variants evaluated on individual stacks.

# Appendix 5

## Supplemental tables

**Appendix 5—table 1.** Ablation study of boundary detection accuracy.

Accuracy of boundary prediction was assessed for twelve training procedures that sample different type of architecture (3D U-Net *vs.* 3D Residual U-Net), loss function (BCE *vs.* Dice *vs.* BCE-Dice) and normalization (Group-Norm *vs.* Batch-Norm). All entries are evaluated at a fix threshold of 0.5. Reported values are the means and standard deviations for a set of seven specimen for the ovules and four for the lateral root primordia. Source files used to create the table are available in the *Appendix 5—table 1—source data 1*.

| Network and resolution | Accuracy (%) | Precision | Recall | F 1 |
|---|---|---|---|---|
| Ovules | | | | |
| 3D-Unet, $L_{bce}$, Group-Norm | 97.9 1.0 | 0.812 0.083 | 0.884 0.029 | 0.843 0.044 |
| 3D-Unet, $L_{bce}$, Batch-Norm | 98.0 1.1 | 0.815 0.084 | 0.892 0.035 | 0.849 0.047 |
| 3D-Unet, $L_{dice}$, Group-Norm | 97.6 1.0 | 0.765 0.104 | 0.905 0.023 | 0.824 0.063 |
| 3D-Unet, $L_{dice}$, Batch-Norm | 97.8 1.1 | 0.794 0.084 | 0.908 0.030 | 0.844 0.048 |
| 3D-Unet, $L_{bce} + L_{dice}$, Group-Norm | 97.8 1.1 | 0.793 0.086 | 0.907 0.026 | 0.843 0.048 |
| 3D-Unet, $L_{bce} + L_{dice}$, Batch-Norm | 97.9 0.9 | 0.800 0.081 | 0.898 0.025 | 0.843 0.041 |
| 3D-Resunet, $L_{bce}$, Group-Norm | 97.9 0.9 | 0.803 0.090 | 0.880 0.021 | 0.837 0.050 |
| 3D-Resunet, $L_{bce}$, Batch-Norm | 97.9 1.0 | 0.811 0.081 | 0.881 0.031 | 0.841 0.042 |
| 3D-Resunet, $L_{dice}$, Group-Norm | 95.9 2.6 | 0.652 0.197 | 0.889 0.016 | 0.730 0.169 |
| 3D-Resunet, $L_{dice}$, Batch-Norm | 97.9 1.1 | 0.804 0.087 | 0.894 0.035 | 0.844 0.051 |
| 3D-Resunet, $L_{bce} + L_{dice}$, Group-Norm | 97.8 1.1 | 0.812 0.085 | 0.875 0.026 | 0.839 0.044 |
| 3D-Resunet, $L_{bce} + L_{dice}$, Batch-Norm | 98.0 1.0 | 0.815 0.087 | 0.892 0.035 | 0.848 0.050 |
| Lateral Root Primordia | | | | |
| 3D-Unet, $L_{bce}$, Group-Norm | 97.1 1.0 | 0.731 0.027 | 0.648 0.105 | 0.684 0.070 |
| 3D-Unet, $L_{bce}$, Batch-Norm | 97.2 1.0 | 0.756 0.029 | 0.637 0.114 | 0.688 0.080 |
| 3D-Unet, $L_{dice}$, Group-Norm | 96.1 1.1 | 0.587 0.116 | 0.729 0.094 | 0.644 0.098 |
| 3D-Unet, $L_{dice}$, Batch-Norm | 97.0 0.9 | 0.685 0.013 | 0.722 0.103 | 0.700 0.056 |
| 3D-Unet, $L_{bce} + L_{dice}$, Group-Norm | 96.9 1.0 | 0.682 0.029 | 0.718 0.095 | 0.698 0.060 |
| 3D-Unet, $L_{bce} + L_{dice}$, Batch-Norm | 97.0 0.8 | 0.696 0.012 | 0.716 0.101 | 0.703 0.055 |
| 3D-Resunet, $L_{bce}$, Group-Norm | 97.3 1.0 | 0.766 0.039 | 0.668 0.089 | 0.712 0.066 |
| 3D-Resunet, $L_{bce}$, Batch-Norm | 97.0 1.1 | 0.751 0.042 | 0.615 0.116 | 0.673 0.086 |
| 3D-Resunet, $L_{dice}$, Group-Norm | 96.5 0.9 | 0.624 0.095 | 0.743 0.092 | 0.674 0.083 |
| 3D-Resunet, $L_{dice}$, Batch-Norm | 97.0 0.9 | 0.694 0.019 | 0.724 0.098 | 0.706 0.055 |
| 3D-Resunet, $L_{bce} + L_{dice}$, Group-Norm | 97.2 1.0 | 0.721 0.048 | 0.735 0.076 | 0.727 0.059 |
| 3D-Resunet, $L_{bce} + L_{dice}$, Batch-Norm | 97.0 0.9 | 0.702 0.024 | 0.703 0.105 | 0.700 0.063 |

The online version of this article includes the following source data for Table Appendix 5—table 1.:

Appendix 5—table 1—Source data 1. Source data for the ablation study of boundary detection accuracy in Source data for the average segmentation accuracy of different segmentation algorithms in *Appendix 5—table 1*.

'pmaps_root' contains evaluation metrics computed on the test set from the Lateral Root dataset, 'pmaps_ovules' contains evaluation metrics computed on the test set from the Ovules dataset, 'fig2_precision_recall.ipynb' is a Jupyter notebook generating the plots.

**Appendix 5—table 2.** Average segmentation accuracy for different segmentation algorithms.

The average is computed from a set of seven specimen for the ovules and four for the lateral root primordia (LRP), while the error is measured by standard deviation. The segmentation is produced by multicut, GASP, mutex watershed (Mutex) and DT watershed (DTWS) clustering strategies. We

additionally report the scores given by the lifted multicut on the LRP dataset. The Metrics used are the Adapted Rand error to asses the overall segmentation quality, the $VOI_{merge}$ and $VOI_{split}$ respectively assessing erroneous merge and splitting events (lower is better for all metrics). Source files used to create the table are available in the *Appendix 5—table 2—source data 1*.

| Segmentation | ARand | $VOI_{split}$ | $VOI_{merge}$ |
|---|---|---|---|
| Ovules | | | |
| DTWS | 0.135 0.036 | 0.585 0.042 | 0.320 0.089 |
| GASP | 0.114 0.059 | 0.357 0.066 | 0.354 0.109 |
| MultiCut | 0.145 0.080 | 0.418 0.069 | 0.429 0.124 |
| Mutex | 0.115 0.059 | 0.359 0.066 | 0.354 0.108 |
| Lateral Root Primordia | | | |
| DTWS | 0.550 0.158 | 1.869 0.174 | 0.159 0.073 |
| GASP | 0.037 0.029 | 0.183 0.059 | 0.237 0.133 |
| MultiCut | 0.037 0.029 | 0.190 0.067 | 0.236 0.128 |
| Lifted Multicut | 0.040 0.039 | 0.162 0.068 | 0.287 0.207 |
| Mutex | 0.105 0.118 | 0.624 0.812 | 0.542 0.614 |

The online version of this article includes the following source data for Table Appendix 5—table 2.:

Appendix 5—table 2—Source data 1. Source data for the average segmentation accuracy of different segmentation algorithms in *Appendix 5—table 2*.

The archive contains CSV files with evaluation metrics computed on the Lateral Root and Ovules test sets.

'root_final_16_03_20_110904.csv' - evaluation metrics for the Lateral Root, 'ovules_final_16_03_20_113546.csv' - evaluation metrics for the Ovules.

**Appendix 5—table 3.** Average segmentation accuracy on leaf surfaces.

The evaluation was computed on six specimen (data available under: https://osf.io/kfx3d) with the segmentation methodology presented in section *Analysis of leaf growth and differentiation*. The Metrics used are: the ARand error to asses the overall segmentation quality, the $VOI_{merge}$ and $VOI_{split}$ assessing erroneous merge and splitting events respectively, and accuracy (Accu.) measured as percentage of correctly segmented cells (lower is better for all metrics except accuracy). For the Proj3D method a limited number of cells (1.04% mean across samples) was missing due to segmentation errors and required manual seeding. While it is not possible to quantify the favorable impact on the ARand and VOIs scores, we can assert that the Proj3D accuracy has been overestimated by approximately 1.04%.

| Segmentation | ARand | $VOI_{split}$ | $VOI_{merge}$ | Accu. (%) | ARand | $VOI_{split}$ | $VOI_{merge}$ | Accu. (%) |
|---|---|---|---|---|---|---|---|---|
| | Sample 1 (Arabidopsis, Col0_07 T1) | | | | Sample 2 (Arabidopsis, Col0_07 T2) | | | |
| PredAutoSeg | 0.387 | 0.195 | 0.385 | 91.561 | 0.269 | 0.171 | 0.388 | 89.798 |
| Proj3D | 0.159 | 0.076 | 0.273 | 82.700 | 0.171 | 0.078 | 0.279 | 84.697 |
| RawAutoSeg | 0.481 | 0.056 | 0.682 | 75.527 | 0.290 | 0.064 | 0.471 | 75.198 |
| | Sample 3 (Arabidopsis, Col0_03 T1) | | | | Sample 4 (Arabidopsis, Col0_03 T2) | | | |
| PredAutoSeg | 0.079 | 0.132 | 0.162 | 90.651 | 0.809 | 0.284 | 0.944 | 90.520 |
| Proj3D | 0.065 | 0.156 | 0.138 | 88.655 | 0.181 | 0.228 | 0.406 | 91.091 |
| RawAutoSeg | 0.361 | 0.101 | 0.412 | 88.130 | 0.295 | 0.231 | 0.530 | 85.037 |
| | Sample 5 (Cardamine, Ox T1) | | | | Sample 6 (Cardamine, Ox T2) | | | |
| PredAutoSeg | 0.087 | 0.162 | 0.125 | 98.858 | 0.052 | 0.083 | 0.077 | 97.093 |
| Proj3D | 0.051 | 0.065 | 0.066 | 95.958 | 0.037 | 0.060 | 0.040 | 98.470 |
| RawAutoSeg | 0.429 | 0.043 | 0.366 | 93.937 | 0.267 | 0.033 | 0.269 | 89.288 |

# Appendix 6

## PlantSeg - Parameters guide

The PlantSeg workflow can be customized and optimized by tuning the pipeline's hyperparameters. The large number of available options can be intimidating for new user, therefore we provide a short guide to explain them. For more detailed and up-to-date guidelines please visit the project's GitHub repository: https://github.com/hci-unihd/plant-seg.

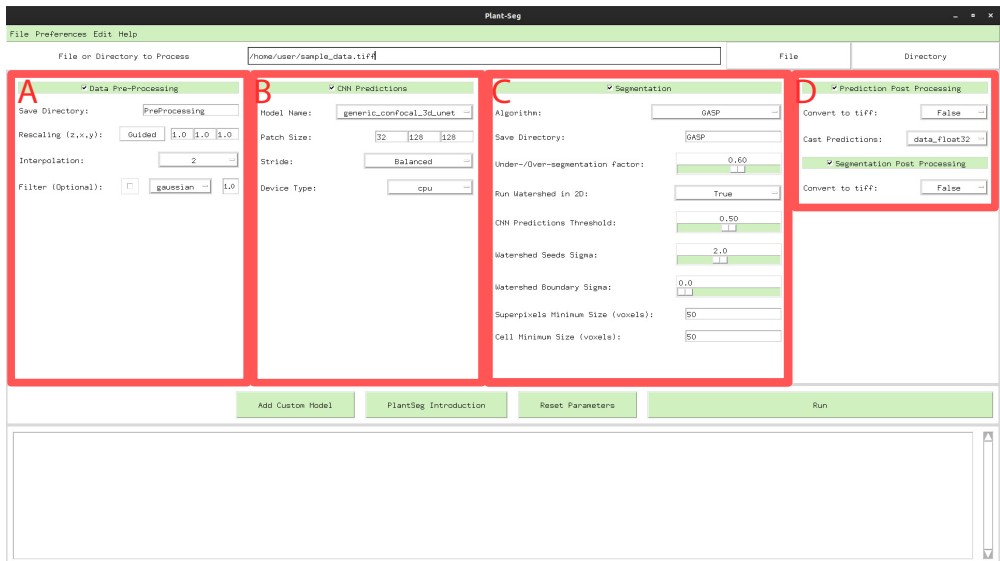

**Appendix 6—figure 1.** PlantSeg GUI. The interface allows to configure all execution steps of the segmentation pipeline, such as: selecting the neural network model and specifying hyperparameters of the partitioning algorithm. *Appendix 6—table 1* describes the Pre-processing (**A**) parameters. *Appendix 6—table 2* provides parameters guide for the CNN Predictions and Post-processing (**B**, **D**). Hyperparameters for Segmentation and Post-processing (**C, D**) are described in *Appendix 6—table 3*.

**Appendix 6—table 1.** Parameters guide for Data Pre-processing.
Menu A in *Figure 1*.

| Process type | Parameter name | Description | Range | Default |
|---|---|---|---|---|
| Data Pre-processing | Save Directory | Create a new sub folder where all results will be stored. | text | 'PreProcessing' |
| | Rescaling (z, y, z) | The rescaling factor can be used to make the data resolution match the resolution of the dataset used in training. Pressing the 'Guided' button in the GUI a widget will help the user setting up the right rescaling | tuple | $[1.0, 1.0, 1.0]$ |
| | Interpolation | Defines order of the spline interpolation. The order 0 is equivalent to nearest neighbour interpolation, one is equivalent to linear interpolation and two quadratic interpolation. | menu | 2 |
| | Filter | Optional: perform Gaussian smoothing or median filtering on the input. Filter has an additional parameter that set the sigma (gaussian) or disc radius (median). | menu | Disabled |

**Appendix 6—table 2.** Parameters guide for CNN Predictions and Post-processing.
Menu B and D in *Appendix 6—figure 1*.

| Process type | Parameter name | Description | Range | Default |
|---|---|---|---|---|
| CNN Prediction | Model Name | Trained model name. Models trained on confocal (model name: 'generic_confocal_3D_unet') and lightsheet (model name: 'generic_confocal_3D_unet') data as well as their multi-resolution variants are available: More info on available models and importing custom models can be found in the project repository. | text | 'generic_confocal…' |
| | Patch Size | Patch size given to the network. A bigger patches cost more memory but can give a slight improvement in performance. For 2D segmentation the Patch size relative to the z axis has to be set to 1. | tuple | [32, 128, 128] |
| | Stride | Specifies the overlap between neighboring patches. The bigger the overlap the the better predictions at the cost of additional computation time. In the GUI the stride values are automatically set, the user can choose between: Accurate (50% overlap between patches), Balanced (25% overlap between patches), Draft (only 5% overlap between patches). | menu | Balanced |
| | Device Type | If a CUDA capable gpu is available and setup correctly, 'cuda' should be used, otherwise one can use 'cpu' for cpu only inference (much slower). | menu | 'cpu' |
| Prediction Post-processing | Convert to tiff | If True the prediction is exported as tiff file. | bool | False |
| | Cast Predictions | Predictions stacks are generated in 'float32'. Or 'uint8' can be alternatively used to reduce the memory footprint. | menu | 'data_float32' |

**Appendix 6—table 3.** Parameters guide for Segmentation. Menu C and D in *Appendix 6—figure 1*.

| Process type | Parameter name | Description | Range | Default |
|---|---|---|---|---|
| Segmentation | Algorithm | Defines which algorithm will be used for segmentation. | menu | 'GASP' |
| | Save Directory | Create a new sub folder where all results will be stored. | text | 'GASP' |
| | Under/Over seg. fac. | Define the tendency of the algorithm to under of over segment the stack. Small value bias the result towards under-segmentation and large towards over-segmentation. | (0.0…1.0) | 0.6 |
| | Run Watersed in 2D | If True the initial superpixels partion will be computed slice by slice, if False in the whole 3D volume at once. While slice by slice drastically improve the speed and memory consumption, the 3D is more accurate. | bool | True |
| | CNN Prediction Threshold | Define the threshold used for superpixels extraction and Distance Transform Watershed. It has a crucial role for the watershed seeds extraction and can be used similarly to the 'Unde/Over segmentation factor' to bias the final result. An high value translate to less seeds being placed (more under segmentation), while with a low value more seeds are placed (more over segmentation). | (0.0…1.0) | 0.5 |
| | Watershed Seeds Sigma | Defines the amount of smoothing applied to the CNN predictions for seeds extraction. If a value of 0.0 used no smoothing is applied. | float | 2.0 |
| | Watershed Boundary Sigma | Defines the amount of Gaussian smoothing applied to the CNN predictions for the seeded watershed segmentation. If a value of 0.0 used no smoothing is applied. | float | 0.0 |
| | Superpixels Minimum Size | Superpixels smaller than the threshold (voxels) will be merged with a the nearest neighbour segment. | integer | 50 |
| | Cell Minimum Size | Cells smaller than the threshold (voxels) will be merged with a the nearest neighbour cell. | integer | 50 |

*Continued on next page*

*Appendix 6—table 3 continued*

| Process type | Parameter name | Description | Range | Default |
|---|---|---|---|---|
| Segmentation Post-processing | Convert to tiff | If True the segmentation is exported as tiff file. | bool | False |

# Appendix 7

## Empirical example of parameter tuning

PlantSeg's default parameters have been chosen to yield the best result on our core datasets (Ovules, LRP). Furthermore, not only segmentation performances but also resource requirements have been taken into account, e.g. super-pixels are extracted in 2D by default in order to reduce the pipeline runtime and memory usage. Nevertheless, when applying PlantSeg on a new dataset tuning parameters tuningthe default parameters can lead to improved segmentation results. Unfortunately, image stacks can vary in: imaging acquisition, voxels resolutions, noise, cell size and cell morphology. All those factors make it very hard to define generic guidelines and optimal results will always require some trial and error.

Here we present an example of empirical parameter tuning where we use a 3D Cell Atlas as a test dataset since we can quantitatively evaluate the pipeline's performance on it. We will show how we can improve on top of the results presented in *Table 1* of the main manuscript. Since the number of parameters does not allow for a complete grid search we will focus only on three key aspects: model/rescaling, over/under segmentation factor and 3D vs 2D super pixels.

- *model/rescaling*: As we showed in the main text, on this particular data the default confocal CNN model already provides a solid performance. If we now take into consideration the voxel resolution, we have two possible choices. Using a CNN model trained on a more similar resolution or if this is not possible using the rescaling factor to reduce the resolution gap further. In *Appendix 7—table 1* we can see how the results vary if we take into consideration those two aspects. In particular we can observe that in this case the rescaling of voxels depth of a factor 3x considerably improved the overall scores.
- *over/under segmentation factor*: Now that we have tuned the network predictions we can move to tuning the segmentation performance. The main parameter we can use is the *over/under segmentation factor*, this will try to compensate the over- or under-segmentation. From the results in *Appendix 7—table 1* we can observe a strong tendency towards under-segmentation, this suggest that increasing the *over/under segmentation factor* will balance the segmentation. In table *Appendix 7—table 2* we can see the results for three different values, increasing the *over/under segmentation factor* as the desired effect and overall improved the results.
- *3D vs 2D super pixels*: Already tuning this two aspects of the pipeline drastically improved the segmentation according to our metrics. as a final tweak we can switch to 3D super pixels to further improve results. In *Appendix 7—table 3* we present the final results. Overall the final improvement is roughly a factor of ×2 in terms of ARand score compared to PlantSeg default.

Further fine tuning could be performed on the PlantSeg parameters to further improve the scores.

**Appendix 7—table 1.** Comparison between the *generic confocal* CNN model (default in PlantSeg), the closest confocal model in terms of xy plant voxels resolution *ds3 confocal* and the combination of *ds3 confocal* and rescaling (in order to mach the training data resolution a rescaling factor of $(3, 1, 1)$ zxy has been used).

The later combination showed the best overall results. To be noted that *ds3 confocal* was trained on almost isotropic data, while the 3D Digital Tissue Atlas is not isotropic. Therefore poor performances without rescaling are expected. Segmentation obtained with GASP and default parameters

| Dataset | Generic confocal (Default) | | | ds3 confocal | | | ds3 confocal + rescaling | | |
|---|---|---|---|---|---|---|---|---|---|
| | ARand | $VOI_{split}$ | $VOI_{merge}$ | ARand | $VOI_{split}$ | $VOI_{merge}$ | ARand | $VOI_{split}$ | $VOI_{merge}$ |
| Anther | 0.328 | 0.778 | 0.688 | 0.344 | 1.407 | 0.735 | 0.265 | 0.748 | 0.650 |
| Filament | 0.576 | 1.001 | 1.378 | 0.563 | 1.559 | 1.244 | 0.232 | 0.608 | 0.601 |
| Leaf | 0.075 | 0.353 | 0.322 | 0.118 | 0.718 | 0.384 | 0.149 | 0.361 | 0.342 |
| Pedicel | 0.400 | 0.787 | 0.869 | 0.395 | 1.447 | 1.082 | 0.402 | 0.807 | 1.161 |
| Root | 0.248 | 0.634 | 0.882 | 0.219 | 1.193 | 0.761 | 0.123 | 0.442 | 0.592 |
| Sepal | 0.527 | 0.746 | 1.032 | 0.503 | 1.293 | 1.281 | 0.713 | 0.652 | 1.615 |

*Continued on next page*

*Appendix 7—table 1 continued*

| Dataset | Generic confocal (Default) | | | ds3 confocal | | | ds3 confocal + rescaling | | |
|---|---|---|---|---|---|---|---|---|---|
| | ARand | $VOI_{split}$ | $VOI_{merge}$ | ARand | $VOI_{split}$ | $VOI_{merge}$ | ARand | $VOI_{split}$ | $VOI_{merge}$ |
| Valve | 0.572 | 0.821 | 1.315 | 0.617 | 1.404 | 1.548 | 0.586 | 0.578 | 1.443 |
| Average | 0.389 | 0.731 | 0.927 | 0.394 | 1.289 | 1.005 | 0.353 | 0.600 | 0.915 |

**Appendix 7—table 2.** Comparison between the results obtained with three different *over/under segmentation factor* (0.5, 0.6, 0.7).
The effect of tuning this parameter is mostly reflected in the VOIs scores. In this case the best result have been obtained by steering the segmentation towards the over segmentation.

**ds3 confocal + rescaling**

| Dataset | Over/under factor 0.5 | | | Over/under factor 0.6 (Default) | | | Over/under factor 0.7 | | |
|---|---|---|---|---|---|---|---|---|---|
| | ARand | $VOI_{split}$ | $VOI_{merge}$ | ARand | $VOI_{split}$ | $VOI_{merge}$ | ARand | $VOI_{split}$ | $VOI_{merge}$ |
| Anther | 0.548 | 0.540 | 1.131 | 0.265 | 0.748 | 0.650 | 0.215 | 1.130 | 0.517 |
| Filament | 0.740 | 0.417 | 1.843 | 0.232 | 0.608 | 0.601 | 0.159 | 0.899 | 0.350 |
| Leaf | 0.326 | 0.281 | 0.825 | 0.149 | 0.361 | 0.342 | 0.117 | 0.502 | 0.247 |
| Pedicel | 0.624 | 0.585 | 2.126 | 0.402 | 0.807 | 1.161 | 0.339 | 1.148 | 0.894 |
| Root | 0.244 | 0.334 | 0.972 | 0.123 | 0.442 | 0.592 | 0.113 | 0.672 | 0.485 |
| Sepal | 0.904 | 0.494 | 2.528 | 0.713 | 0.652 | 1.615 | 0.346 | 0.926 | 1.211 |
| Valve | 0.831 | 0.432 | 2.207 | 0.586 | 0.578 | 1.443 | 0.444 | 0.828 | 1.138 |
| Average | 0.602 | 0.441 | 1.662 | 0.353 | 0.600 | 0.915 | 0.248 | 0.872 | 0.691 |

**Appendix 7—table 3.** Comparison between 2D vs 3D super pixels.
From out experiments, segmentation quality is almost always improved by the usage of 3D super pixels. On the other side, the user should be aware that this improvement comes at the cost of a large slow-down of the pipeline (roughly × 4.5 on our system Intel Xenon E5-2660, RAM 252 Gb).

**ds3 confocal + rescaling**

**Over/under factor 0.7**

| Dataset | Super Pixels 2D (Default) | | | | Super Pixels 3D | | | |
|---|---|---|---|---|---|---|---|---|
| | ARand | $VOI_{split}$ | $VOI_{merge}$ | time (s) | ARand | $VOI_{split}$ | $VOI_{merge}$ | time (s) |
| Anther | 0.215 | 1.130 | 0.517 | 600 | 0.167 | 0.787 | 0.399 | 2310 |
| Filament | 0.159 | 0.899 | 0.350 | 120 | 0.171 | 0.687 | 0.487 | 520 |
| Leaf | 0.117 | 0.502 | 0.247 | 800 | 0.080 | 0.308 | 0.220 | 3650 |
| Pedicel | 0.339 | 1.148 | 0.894 | 450 | 0.314 | 0.845 | 0.604 | 2120 |
| Root | 0.113 | 0.672 | 0.485 | 210 | 0.101 | 0.356 | 0.412 | 920 |
| Sepal | 0.346 | 0.926 | 1.211 | 770 | 0.257 | 0.690 | 0.966 | 3420 |
| Valve | 0.444 | 0.828 | 1.138 | 530 | 0.300 | 0.494 | 0.875 | 2560 |
| Average | 0.248 | 0.872 | 0.691 | 500 | 0.199 | 0.595 | 0.566 | 2210 |

