## [Decision Letter]

**Acceptance summary:**

In this manuscript we are introduced to PlantSeg, an automated, versatile new image analysis pipeline with features that work especially well on plant tissues. PlantSeg uses machine learning (convolutional neural networks) to identify cellular boundaries and to segment complex tissues into their constituent cells. This pipeline performs well over a range of different tissues, is accessible to novice and expert users, and can be combined with other software to enable quantitative assessment of biological features.

**Decision letter after peer review:**

Thank you for submitting your article "Accurate and versatile 3D segmentation of plant tissues at cellular resolution" for consideration by *eLife*. Your article has been reviewed by three peer reviewers, including Dominique C Bergmann as the Reviewing Editor and Reviewer #1, and the evaluation has been overseen by Christian Hardtke as the Senior Editor. The following individual involved in review of your submission has agreed to reveal their identity: Moritz Graeff (Reviewer #3).

The reviewers have discussed the reviews with one another and the Reviewing Editor has drafted this decision to help you prepare a revised submission.

Summary:

PlantSeg, a tool described in Wolny et al.: "Accurate and versatile 3D segmentation of plant tissues at cellular resolution", harnesses recent advances in computer vision to perform volumetric segmentation of plant tissues. It describes an image analysis pipeline utilising machine learning for cellular segmentation of plant tissues, an essential but often time-consuming part of many recent studies in plant development. The authors provide an automated, versatile new pipeline that utilises convolutional neural networks to identify cellular boundaries before cell segmentation. This pipeline performs well over a range of different tissues, is accessible to novice and expert users, and will likely attract great interest within the plant (and animal) scientific community

Three reviewers concluded that this tool addressed a pressing need for the community, and in general, found the design, rationale, and performance good. There were several suggestions to improve these points as well as to improve the ease of installation and use. Below, we divide essential revisions into two categories: performance and usability, and these should be addressed in a revision and the response to reviewers.

Essential revisions:

Performance:

1) The authors compare nine different CNN designs, varying in the network architecture, loss function, and training protocol (including image augmentation and changes of layer orders within the training protocol). It is unclear how the tested designs were chosen out of the large number of possible design permutations. In particular, it is unclear why the authors do not include further design permutations using the design they consider the most robust (based on the Lbce + Ldice loss function) – only a single design uses this loss function, although it can reasonably be expected that other designs using this function might further improve CNN performance. Can the authors either include such networks, or explain why they chose not to, and furthermore lay out a clear rationale for choosing the networks presented here?

PlantSeg is accessible to non-experts, but its exact advantages over other user-friendly tools such as the U-Net ImageJ plug-in, CDeep3M, or ilastik could use further elaboration. Relative to these methods, what are PlantSeg's primary contributions-that it combines CNN-based predictions and more sophisticated post-processing methods for plant tissues?

2) In the sections assessing the performance of PlantSeg in different datasets, the authors do not specify which CNN they used for boundary detection, and which segmentation strategy they used. Did they preselect their strategy based on microscope type and voxel size, as recommended later on for other users? Or did they test multiple combinations and identified the best performing one? Considering it is not feasible to generate a ground truth for every dataset analysed, it is of great interest to the reader to understand the range in performance of the different CNN/segmentation combinations available in the PlantSeg pipeline. If the authors tested multiple combinations, they should report their results. If they used a single combination, they should explain how this was chosen.

3) In the subsection "Analysis of leaf growth and differentiation", the authors specify the mean number of segmentation errors to assess the quality of the PlantSeg pipeline compared to MorphoGraphX. It is not clear how the ground truth for these comparisons was generated, and also, why the authors deviate from their more detailed assessment of segmentation quality used before (subsections "Step 2: segmentation of tissues into cells using graph partitioning" and "Performance on external plant datasets").

4) It is unclear how PlantSeg perform on one of the most characteristic (and problematic) cell types-the highly lobed epidermal cells. Lobes have presented challenges to older watershed-based algorithms. The runs of PlantSeq on sepals, which contain these cells do not seem to have behaved well (likely due to the low quality of the input data) and the work on Cardamine leaves appears to be a combination of MorphoGraphX and PlantSeg. There are other published datasets that include lobed cells from Arabidopsis leaves and also maize leaves (where lobing is of a different nature) and these should be analyzed with PlantSeg alone to demonstrate its effectiveness at segmenting such cells.

Usability:

5) This is a Linux-based program, and this diminishes its usability especially for people who might want to use it at home on iOS/PC systems during the current pandemic. While we recognize that changing the structure to run on these systems is a big request and it is not an absolute requirement for this manuscript to be accepted, it is something that needs to be acknowledged. Writing early in the text (even in the Abstract) that this is Linux-based should cue in the reader about requirements.

6) To generate a tool that is both accurate and generalizable, you experimented with several design choices, including the network architecture, loss function, patch size, order of operations within a U-Net level, and partitioning strategy. The pre-trained networks are included in the software package and can be specified via the graphical user interface (GUI) or the command line. Here, non-experts would benefit from more guidance as to which pre-trained networks they should specify for which datasets. For example, beyond considerations such as microscope modality and voxel size, what are the guiding principles for which partitioning strategy should be selected? If this information is already available, please refer to its location in the main text.

7) Additionally, the GUI allows users to adjust a number of parameters. To expand the userbase, consider providing an appendix that (1) explains what these parameters mean and (2) outlines the circumstances under which they should be adjusted.

8) A valuable addition would be a table that lists how PlantSeg interfaces with other image analysis tools, specifically including software packages (in addition to MorphoGraphX) that can perform cell counting, cell tracking, and cell volume and shape measurements on the outputs of PlantSeg.

9) Finally, in the GitHub repository, the read-me document is helpful, but the folders and files are not named in an intuitive way for non-experts to navigate. Please rename and/or provide a short description so it is clear what each folder contains.

---

## [Author Response]

Essential revisions:Performance:1) The authors compare nine different CNN designs, varying in the network architecture, loss function, and training protocol (including image augmentation and changes of layer orders within the training protocol). It is unclear how the tested designs were chosen out of the large number of possible design permutations. In particular, it is unclear why the authors do not include further design permutations using the design they consider the most robust (based on the Lbce + Ldice loss function) – only a single design uses this loss function, although it can reasonably be expected that other designs using this function might further improve CNN performance. Can the authors either include such networks, or explain why they chose not to, and furthermore lay out a clear rationale for choosing the networks presented here?

We thank the reviewers for pointing out the missing reasoning behind the particular network design choices used in our work. The rationale for choosing those particular aspects of neural network training is now described in the subsection “Step 1: cell boundary detection” which says:

“Aiming for the best performance across different microscope modalities, we explored various components of neural network design critical for improved generalization and robustness to noise, namely: the network architecture, loss function, normalization layers and size of patches used for training”.

Additionally we’ve also updated the Appendix 5—table 1 in which our complete grid search exploration is now presented, where we compare 2 different architecture variants, 3 different loss functions and 2 different normalization/patch size specifications. Clearly, we have not explored all possible design choices, but only the most popular variants. For network architecture in particular, other block or layer designs could be used, and networks other than U-nets have been introduced in the literature. However, the U-net remains the method of choice for microscopy images, it is widely used, robust and easy to train. We believe our non-expert users will benefit more from a solid implementation of the well-studied state-of-the-art network. Still, to make PlantSeg future-proof, we allow users to supply their own pre-trained networks and we will ourselves monitor the field and add new champion networks to PlantSeg once they emerge in the bioimage computing community.

PlantSeg is accessible to non-experts, but its exact advantages over other user-friendly tools such as the U-Net ImageJ plug-in, CDeep3M, or ilastik could use further elaboration. Relative to these methods, what are PlantSeg's primary contributions-that it combines CNN-based predictions and more sophisticated post-processing methods for plant tissues?

Regarding the PlantSeg’s primary contributions relative to other tools (ImageJ’s U-Net plugin, CDeep3M), we have added the following to the Introduction section:

“Combining the repository of state-of-the-art neural networks trained on the two common microscope modalities and going beyond just thresholding or watershed with robust graph partitioning strategies is the main strength of our package.”

In more in detail:

We differ from ImageJ’s U-Net plugin mainly by the segmentation approach. Our robust graph partitioning is especially well-suited for cell segmentation in densely packed tissue. In contrast the ImageJ’s U-Net plugin relies on a background/foreground segmentation approach that is mostly suited for more sparsely placed cells, as no advanced post-processing is made available.

CDeep3M main target application is electron microscopy images. Both the neural network architecture and the pre-trained models are optimized for this modality. Therefore, it cannot be used in the context of confocal and light sheet microscopy without re-training the models.

Moreover, we provide a tool that can be easily installed on local hardware, while CDeep3M relies on the Amazon cloud infrastructure.

The current beta version of ilastik supports inference with neural networks. However, this version is not in production yet and, since the scope of ilastik is much larger than the scope of PlantSeg, it would take a while until this part reaches full maturity, especially for training (note that ilastik is developed in Anna Kreshuk's lab at EMBL, so we are fully aware of the internal planning of its future). While we do envision eventual incorporation of the PlantSeg functionality into ilastik, we believe PlantSeg has a lot of value as a standalone product that does one thing and does it well.

2) In the sections assessing the performance of PlantSeg in different datasets, the authors do not specify which CNN they used for boundary detection, and which segmentation strategy they used. Did they preselect their strategy based on microscope type and voxel size, as recommended later on for other users? Or did they test multiple combinations and identified the best performing one? Considering it is not feasible to generate a ground truth for every dataset analysed, it is of great interest to the reader to understand the range in performance of the different CNN/segmentation combinations available in the PlantSeg pipeline. If the authors tested multiple combinations, they should report their results. If they used a single combination, they should explain how this was chosen.

We agree with the reviewer that the parameters choice should have been reported in the manuscript. We now mention the relevant parameters in each caption where quantitative results are shown.

The results presented in the manuscript for the performance of PlantSeg in different datasets are mostly obtained with default parameters which were chosen by quantitative evaluation on the Ovules and LRP datasets. Our aim was to show the performance of PlantSeg out of the box, before more advanced parameter tuning.

As pointed out by the reviewers PlantSeg offers a large number of options for fine-tuning the pipeline result. This freedom makes it very challenging to find shared parameters that are optimal for all datasets presented. Moreover, the variety of voxels resolutions, noise, cell size and cell morphology makes it very hard to define truly generic guidelines, and optimal results will always require some trial and error. These observations are reported in the appendix “Empirical Example of Parameter Tuning”, the paragraph reads:

“PlantSeg's default parameters have been chosen to yield the best result on our core datasets (Ovules, LRP). […] Unfortunately, image stacks can vary in: imaging acquisition, voxels resolutions, noise, cell size and cell morphology. All those factors make it very hard to define generic guidelines and optimal results will always require some trial and error.”

To illustrate our guidelines, we added an empirical example of how to tune PlantSeg for a particular dataset in the appendix “Empirical Example of Parameter Tuning”. Here we show how to tune PlantSeg parameters in three simple steps. The improvement can be noticed qualitatively, but we also evaluate it quantitatively on the Arabidopsis 3D cell atlas dataset.

3) In the subsection "Analysis of leaf growth and differentiation", the authors specify the mean number of segmentation errors to assess the quality of the PlantSeg pipeline compared to MorphoGraphX. It is not clear how the ground truth for these comparisons was generated, and also, why the authors deviate from their more detailed assessment of segmentation quality used before (subsections "Step 2: segmentation of tissues into cells using graph partitioning" and "Performance on external plant datasets").

We updated the main text to make it more clear why surface segmentations and MorphoGraphX were used and how PlantSeg can assist with creating surface segmentations. Furthermore, we extended the segmentation quality assessment by the previously introduced quality measures ARand, VOI_split_ and VOI_merge_. Also, we revisited the methods to state more explicitly how the ground truth was generated. This part now reads as follows:

“To track leaf growth, the same sample is imaged over the course of several days, covering a volume much larger than ovules or meristems. […] In contrast, the RawAutoSeg method achieved the poorest mean scores in both measures demonstrating that using PlantSeg improved the surface segmentation (see Figure 9).”

In addition, a more complete explanation of the leaf surface workflow and ground truth generation has been added in the new subsection “Creation of leaf surface segmentations”. In particular the new text regarding the ground truth generation now reads as follows:

“To compare the segmentations created by MorphoGraphX alone with the ones using PlantSeg’s files as input, we first obtained a ground-truth segmentation using the MorphographX auto-segmentation pipeline as described in Strauss et al., 2019, (Figure 10B) and manually fixed all segmentation errors using processes in MorphoGraphX.”

4) It is unclear how PlantSeg perform on one of the most characteristic (and problematic) cell types-the highly lobed epidermal cells. Lobes have presented challenges to older watershed-based algorithms. The runs of PlantSeq on sepals, which contain these cells do not seem to have behaved well (likely due to the low quality of the input data) and the work on Cardamine leaves appears to be a combination of MorphoGraphX and PlantSeg. There are other published datasets that include lobed cells from Arabidopsis leaves and also maize leaves (where lobing is of a different nature) and these should be analyzed with PlantSeg alone to demonstrate its effectiveness at segmenting such cells.

We thank the reviewers for pointing out these published datasets. We fully agree that highly lobed cells make for an interesting use case and we have now evaluated Plant Seg on the data from Foxe et al. [1].

The results are presented in subsection: “Performance on external plant datasets”. We included a new figure to show them qualitatively and updated the text of the section as follows:

“Figure 5. Qualitative results on the highly lobed epidermal cells from Fox et al., 2018. […] In order to show pre-trained networks' ability to generalize to external data, we additionally depict PlantSeg's boundary predictions (third row, middle).”

We have also changed the text in subsection “Performance on external plants dataset” accordingly to the additional experiments. The section now reads as follow:

“To test the generalization capacity of PlantSeg, we assessed its performance on data for which no network training was performed. […] This demonstrates the generalization capacity of the pre-trained models from the PlantSeg package.”

Usability:5) This is a Linux-based program, and this diminishes its usability especially for people who might want to use it at home on iOS/PC systems during the current pandemic. While we recognize that changing the structure to run on these systems is a big request and it is not an absolute requirement for this manuscript to be accepted, it is something that needs to be acknowledged. Writing early in the text (even in the Abstract) that this is Linux-based should cue in the reader about requirements.

We thank the reviewers for pointing out all of the PlantSeg usability issues found in the manuscript. Regarding the support of other operating systems: both MacOS and Windows 10 are supported via the docker images containing the PlantSeg package. The instructions of how to run docker images on MacOS and Windows 10 are included in the project README page on GitHub (https://github.com/hci-unihd/plant-seg#docker-image).

In order to improve the usability of the package even more we’re currently working towards releasing the binaries for Windows 10 and MacOS by the end of the year, so that no additional software (in this case the Docker engine) will be necessary to run PlantSeg on Windows and MacOS.

6) To generate a tool that is both accurate and generalizable, you experimented with several design choices, including the network architecture, loss function, patch size, order of operations within a U-Net level, and partitioning strategy. The pre-trained networks are included in the software package and can be specified via the graphical user interface (GUI) or the command line. Here, non-experts would benefit from more guidance as to which pre-trained networks they should specify for which datasets. For example, beyond considerations such as microscope modality and voxel size, what are the guiding principles for which partitioning strategy should be selected? If this information is already available, please refer to its location in the main text.

We thank the reviewers for pointing out the lack of clear explanation of how the appropriate network and the partitioning strategy can be chosen by the user.

Regarding the choice of the network, the voxel size and microscope modality are the only and most critical parameters which have to be specified by the user. This fact has been already explained in the subsection “A package for plant tissue segmentation and benchmarking”:

“Users can select from the available set of pre-trained networks the one with features most similar to their datasets. Alternatively, users can let PlantSeg select the pre-trained network based on the microscope modality (light sheet or confocal) and voxel size”.

The specific architectural design of the pre-trained networks are hidden from the user. Those were only relevant in the exploratory phase where we looked for the right combination of the network architecture, loss function and the normalization layers that would work best on our two core datasets. After finding the best combination of those design choices, we fixed them and trained separate networks for different microscope modalities and voxel sizes.

With regard to the partitioning strategy, in the subsection “Segmentation using graph partitioning” we now also explain how to choose the it most effectively:

“As a general guideline for choosing the partitioning strategy on a new data is to start with the GASP algorithm, which is the most generic. […] Importantly the newly proof-read results can then be used to train a better network that can be applied to this type of data in the future (see the appendix \nameref{sec:ground_truth_creation} for an overview of this process).”

7) Additionally, the GUI allows users to adjust a number of parameters. To expand the userbase, consider providing an appendix that (1) explains what these parameters mean and (2) outlines the circumstances under which they should be adjusted.

We agree that including a parameters guide in the manuscript appendix would greatly help new users. The newly added Appendix 6 contains description of all main parameters available in PlantSeg, while Appendix 7 provides an example of parameter tuning for a particular dataset from the 3D Tissue Atlas.

8) A valuable addition would be a table that lists how PlantSeg interfaces with other image analysis tools, specifically including software packages (in addition to MorphoGraphX) that can perform cell counting, cell tracking, and cell volume and shape measurements on the outputs of PlantSeg.

We agree with the reviewers that since PlantSeg was used in combination with e.g. Paintera for proofreading and MorphographX for mesh extraction and surface segmentation including an explanation of how to integrate PlantSeg with other tools would be helpful for the end users. In the subsection “A package for plant tissue segmentation and benchmarking” we explain that results from PlantSeg can easily be loaded into other tools for further processing of the segmentation results:

“Our software can export both the segmentation results and the boundary probability maps as Hierarchical Data Format (HDF5) or Tagged Image File Format (TIFF). […] In exported boundary probability maps each pixel has a floating point number between 0 and 1 reflecting a probability of that pixel belonging to a cell boundary.”

9) Finally, in the GitHub repository, the read-me document is helpful, but the folders and files are not named in an intuitive way for non-experts to navigate. Please rename and/or provide a short description so it is clear what each folder contains.

The naming conventions used in the repository follow a standard structure for Python packages (described in: https://docs.python-guide.org/writing/structure). Nevertheless, we agree that it can be hard to navigate, so we now included an index with descriptions in the repository’s README page (https://github.com/hci-unihd/plant-seg/blob/master/README.md#repository-index).